# Stress-response balance drives the evolution of a network module and its host genome

Caleb González[1,†], Joe Christian J Ray[1,2,†], Michael Manhart[3,4], Rhys M Adams[1], Dmitry Nevozhay[1,5], Alexandre V Morozov[3,6] & Gábor Balázsi[1,7,8,*]

## Abstract

Stress response genes and their regulators form networks that underlie drug resistance. These networks often have an inherent tradeoff: their expression is costly in the absence of stress, but beneficial in stress. They can quickly emerge in the genomes of infectious microbes and cancer cells, protecting them from treatment. Yet, the evolution of stress resistance networks is not well understood. Here, we use a two-component synthetic gene circuit integrated into the budding yeast genome to model experimentally the adaptation of a stress response module and its host genome in three different scenarios. In agreement with computational predictions, we find that: (i) intra-module mutations target and eliminate the module if it confers only cost without any benefit to the cell; (ii) intra- and extra-module mutations jointly activate the module if it is potentially beneficial and confers no cost; and (iii) a few specific mutations repeatedly fine-tune the module's noisy response if it has excessive costs and/or insufficient benefits. Overall, these findings reveal how the timing and mechanisms of stress response network evolution depend on the environment.

**Keywords** drug resistance; experimental evolution; positive feedback; synthetic gene circuit; tradeoff

**Subject Categories** Quantitative Biology & Dynamical Systems; Synthetic Biology & Biotechnology; Evolution

**Mol Syst Biol. (2015) 11: 827**

## Introduction

The number of human-designed biological systems has increased rapidly since the inception of synthetic biology (Purnick & Weiss, 2009). Parts and concepts underlying synthetic biological constructs

have expanded quickly, feeding on general biological knowledge. Conversely, synthetic biology has enormous but unexploited potential to inform other areas of biology, such as evolutionary biology (Tanouchi et al, 2012b).

For example, gene regulatory networks that control the expression of stress-protective genes have emerged through evolution (Lopez-Maury et al, 2008) but can also be built de novo (Nevozhay et al, 2012; Tanouchi et al, 2012a). Depending on the details of gene regulation, cells can survive because they respond to stress (Gasch et al, 2000); diversify non-genetically (hedge bets), independent of the stress (Balaban et al, 2004; Thattai & van Oudenaarden, 2004; Levy et al, 2012); or use a mixture of these two strategies (New et al, 2014). However, stress-protective gene expression can be costly or toxic in the absence of stress (Andersson & Levin, 1999), or even in the presence of stress when the expression level exceeds the requirement for survival (Nevozhay et al, 2012). Overall, the costs and benefits of survival mechanisms create a tradeoff between maximizing growth while also ensuring survival during stress. How mutations alter stress response networks to improve fitness under such circumstances, especially in phenotypically heterogeneous populations (Sumner & Avery, 2002), is an open problem in evolutionary biology.

Consider a stress response network module, consisting of a stress-sensing transcriptional regulator and its stress-protective gene target, which has arisen in a cell's genome. Similar modules, such as Tn10 (Hillen & Berens, 1994), toxin-antitoxin systems (Yamaguchi et al, 2011), or bypass signaling (Hsieh & Moasser, 2007), can arise rapidly by recombination, horizontal gene transfer, or inhibitor-mediated alternate pathway activation. Considering their impact on microbial and cancer drug resistance, it is important to know how reproducibly and how quickly such stress defense networks can adapt (Lobkovsky & Koonin, 2012). Yet, we currently lack quantitative, hypothesis-driven understanding of how initially suboptimal stress defense modules evolve inside the host genome, especially in the presence of gene expression noise (Balázsi et al, 2011; Munsky et al, 2012; Sanchez & Golding, 2013). Although

1  Department of Systems Biology - Unit 950, The University of Texas MD Anderson Cancer Center, Houston, TX, USA
2  Center for Computational Biology & Department of Molecular Biosciences, University of Kansas, Lawrence, KS, USA
3  Department of Physics & Astronomy, Rutgers University, Piscataway, NJ, USA
4  Department of Chemistry and Chemical Biology, Harvard University, Cambridge, MA, USA
5  School of Biomedicine, Far Eastern Federal University, Vladivostok, Russia
6  BioMaPS Institute for Quantitative Biology, Rutgers University, Piscataway, NJ, USA
7  Laufer Center for Physical & Quantitative Biology, Stony Brook University, Stony Brook, NY, USA
8  Department of Biomedical Engineering, Stony Brook University, Stony Brook, NY, USA
   *Corresponding author. Tel: +1 631 632 5414; Fax: +1 631 632 5405; E-mail: gabor.balazsi@stonybrook.edu
   †These authors contributed equally to this study

network evolution theory (Kauffman, 1993; Mason *et al*, 2004; Kashtan & Alon, 2005) and laboratory evolution experiments (Lenski & Travisano, 1994; Beaumont *et al*, 2009; Tenaillon *et al*, 2012; Toprak *et al*, 2012; Lang *et al*, 2013) have generated important insights, they have provided largely descriptive, *a posteriori* interpretations. Now there is a growing need for predictive, hypothesis-driven, quantitative understanding of gene network evolution, which requires making *a priori* predictions of mutation effects and evolutionary dynamics that are tested experimentally (Wang *et al*, 2013). One option could be to study the evolution of small natural regulatory modules (Dekel & Alon, 2005; Hsu *et al*, 2012; Quan *et al*, 2012; van Ditmarsch *et al*, 2013). However, connections of natural regulatory modules with the rest of the genome can be significant (Maynard *et al*, 2010) and poorly characterized, thus making predictive, quantitative understanding difficult. Synthetic gene circuits (Elowitz & Leibler, 2000; Gardner *et al*, 2000; Stricker *et al*, 2008; Moon *et al*, 2012; Nevozhay *et al*, 2013) represent a better alternative, since they are small, consist of well-characterized components, and typically lack direct regulatory interactions with the host genome. However, it is unclear whether the evolution of synthetic gene circuits (Yokobayashi *et al*, 2002; Sleight *et al*, 2010; Poelwijk *et al*, 2011; Wu *et al*, 2014) can be predicted *a priori*, especially with regard to gene expression heterogeneity.

We recently characterized the dynamics and fitness effects of gene expression for a synthetic two-gene "positive feedback" (PF) circuit (Fig 1A) integrated into the genome of the haploid single-celled eukaryote *Saccharomyces cerevisiae* (Nevozhay *et al*, 2012). This synthetic gene circuit consists of a well-characterized transcriptional regulator (*rtTA*) and an antibiotic resistance gene (*yEGFP::zeoR*). In the presence of tetracycline-analog inducers such as doxycycline, rtTA activates both itself and *yEGFP::zeoR* by binding to two *tetO2* operator sites in two identical promoters (Fig 1A). This positive feedback is noisy, however, and thus, only a fraction of cells switch to high expression of rtTA and yEGFP::zeoR. These cells benefit from high gene expression, which protects them from the antibiotic zeocin. Meanwhile, the same cells experience a cost from rtTA activator expression toxicity, causing a tradeoff when zeocin is present (Nevozhay *et al*, 2012). The fitness (division rate) of any individual cell is the product of its rtTA expression cost and yEGFP:: zeoR expression benefit (Nevozhay *et al*, 2012), which varies from cell to cell. Thus, quantitative knowledge of dynamics and fitness effects makes the PF gene circuit an excellent model for studying gene network evolution in tradeoff situations. Its design separates stress (zeocin) from its adjustable cellular response (inducible *yEGFP::zeoR* expression), facilitating predictive, quantitative understanding of how a stress response module adapts inside the host genome.

Here, we used our quantitative knowledge of the PF gene circuit to predict *a priori* the timing and mechanisms of its initial adaptation to several constant environments (squares in Fig 1B) corresponding to various stress-response imbalance scenarios. We tested these predictions with experimental evolution, followed by sequencing to identify the mutations that establish in the population, depending on the imbalance between the environmental stress and the intracellular response. In this way, we tested how different mutations can readjust the response of a network module with inherent tradeoff, to match the stress and minimize the cost in each specific environment. These results could help us understand how fast and through

what mechanisms drug resistance emerges or deteriorates in the process of network evolution, and could help the future design of synthetic gene circuits that resist evolutionary degradation.

# Results

## The PF gene circuit can mimic various scenarios of stress-response imbalance

We considered the following disparities between the external stress and the activity of a stress defense module: (i) the module responds gratuitously to a harmless environmental change; (ii) the module cannot respond to harmful stress when needed; and (iii) the module responds to stress, but suboptimally. To mimic these scenarios using the PF gene circuit in yeast, we relied on the separability of stress and response, adjusting two environmental factors with known fitness effects (Nevozhay *et al*, 2012): inducer doxycycline and antibiotic zeocin (Fig 1). Hereafter, DxZy will denote environmental conditions, with x and y indicating doxycycline and zeocin concentrations, respectively. The antibacterial compound doxycycline has negligible effect on yeast (Wishart *et al*, 2005), but causes squelching toxicity in engineered PF cells when bound to rtTA (Gari *et al*, 1997; Nevozhay *et al*, 2012). Zeocin is a broad-spectrum DNA-damaging antibiotic (Burger, 1998) that acts on bacteria and eukaryotes.

First, the presence of inducer doxycycline alone corresponds to scenario (i): costly, futile response of some (Fig 1B, DiZ0) or most (Fig 1B, D2Z0) cells that start expressing the PF genes. The cost of response slows the cell division rate of responding, high expressor cells compared to non-responding, low expressor cells (Nevozhay *et al*, 2012). Consequently, the division rate of individual yeast cells can differ drastically from the overall population growth rate. To capture these differences between single cell- and population growth rates, we constructed a population fitness landscape (three-dimensional gray surface in Fig 1B) and cellular fitness landscapes (colored panels in Fig 1B). The population fitness landscape maps the overall population growth against the two environmental variables, doxycycline and zeocin concentrations. Cellular fitness landscapes depict the division rate of single cells versus their gene expression level in a given combination of doxycycline and zeocin. As described in the Appendix, we inferred these landscapes directly from growth rate and gene expression measurements (Appendix Fig S1A) in 13 different combinations of doxycycline and zeocin.

Second, the presence of antibiotic zeocin alone (Fig 1B, D0Z2) corresponds to the lack of response when needed, as in scenario (ii). Finally, the presence of both inducer and antibiotic (Fig 1B, DiZ2 and D2Z2) corresponds to scenario (iii) where the fraction of responding, slower-growing cells ensures cell population survival during antibiotic treatment, but the response is in general suboptimal.

Altogether, the PF gene circuit is a well-characterized module lacking direct regulatory interactions with the yeast genome. It exemplifies typical tradeoffs between the benefits and costs of gene expression in stress response networks. Importantly, the benefits and costs are independently tunable for the PF gene circuit, making it possible to predict and test their evolution toward optimality.

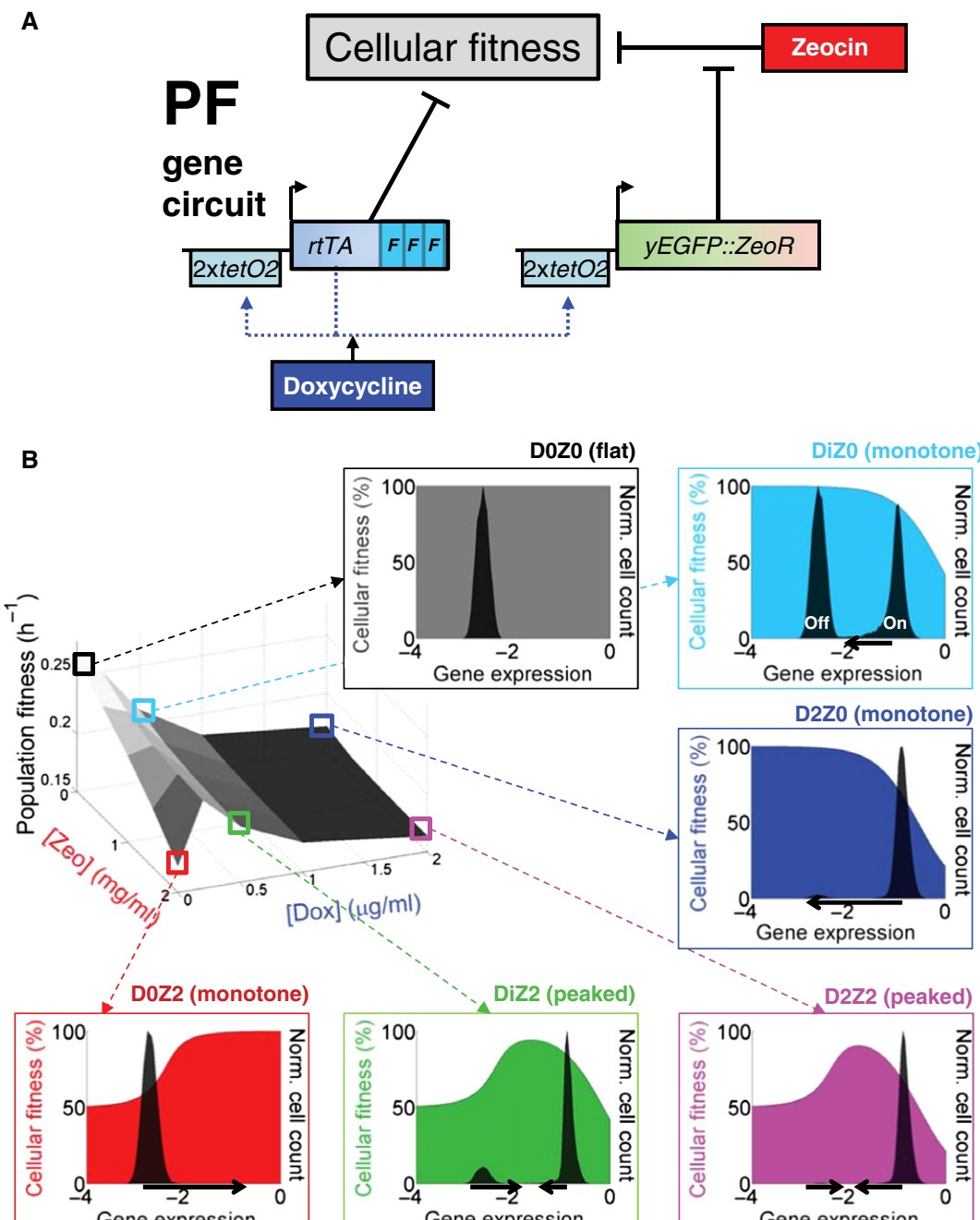

**Figure 1.  The PF synthetic gene circuit: fitness and gene expression characteristics.**

A   The PF synthetic gene circuit (Nevozhay *et al*, 2012) consists of two components. First, the *regulator* reverse tet-trans-activator (rtTA) (Urlinger *et al*, 2000) is a reverse-*tetR* gene fused to three F activator domains (cyan rectangles), which are shorter versions of the VP16 activator (Baron *et al*, 1997). The *target gene* *yEGFP::ZeoR* consists of the fluorescent reporter *yEGFP* fused to the drug resistance gene *zeoR* (Gatignol *et al*, 1988) that binds and inactivates zeocin, a bleomycin-family antibiotic. Unbound zeocin generates DNA double-strand breaks, causing cell cycle arrest and potentially cell death. Doxycycline added to the growth medium diffuses freely through the cell wall and binds to rtTA dimers. Inducer-bound rtTA undergoes a conformational change that results in strong association with two *tetO2* operator sites upstream of each of the two *tetreg* promoters (Becskei *et al*, 2001), activating both regulator and target gene expression, while causing toxicity by squelching.

B   Costs and benefits of PF gene circuit components were determined by measuring cell population growth rate (population fitness) versus two environmental factors: inducer doxycycline and antibiotic zeocin. Each point on the population fitness landscape (three-dimensional gray surface on the left) is an average of cellular fitness values (color-shaded slopes in the surrounding plots) as cells stochastically move within gene expression distributions (black histograms in the surrounding plots). Gene expression is measured as $\log_{10}$(fluorescence) (arbitrary units). DxZy denotes the environment (the x and y following D and Z indicate µg/ml doxycycline and mg/ml zeocin concentrations, respectively, with Di = 0.2 µg/ml doxycycline). Cellular fitness (cell division rate) is a function of gene expression for each cell in each environment DxZy. It is inferred from the population fitness, based on a biochemical model (Nevozhay *et al*, 2012); see the Appendix. The black arrows beneath cellular fitness landscapes illustrate selection pressures pushing the gene expression distribution toward higher fitness.

## Predicting the first evolutionary steps in constant environments

We asked whether the PF cellular and population fitness landscapes (colored squares and panels in Fig 1B; Appendix Fig S1A; Appendix Table S1) could predict evolutionary trends in specific environments. For example, in the D2Z0 environment, most cells are far from their fitness maximum, which is at low expression. If a mutation could push cells downward in expression, toward their fitness maximum (horizontal arrow in Fig 1B, D2Z0 panel), then they should grow faster. Mutations that either *abolish* or *weaken* rtTA toxicity could achieve this effect. Let us call these mutation types "knockout" (K) and "tweaking" (T) mutations, respectively (Fig 2A; Appendix Fig S1B). On the other hand, in the D0Z2 environment cells should benefit from mutations that diminish the effect of the antibiotic. This could happen in various ways, for example by upregulation of native stress-response mechanisms; or by increasing yEGFP::zeoR expression. Let us call these latter mutation types "generic" (G) drug resistance mutations (Fig 2A; Appendix Fig S1B). In all these cases, mutant cells can improve their fitness by unidirectionally lowering or increasing PF gene expression. However, in certain conditions (such as DiZ2), when the cells form two subpopulations that flank the cellular fitness peak, a single-directional expression change is not optimal. This is because a one-way expression shift can only move one subpopulation toward the fitness peak, while the other subpopulation must necessarily move away from it. Instead, optimally the two subpopulations should approach each other, both moving toward the fitness peak (horizontal arrows in Fig 1B, DiZ2, D2Z2 panels).

How would the PF cells evolve to adapt in specific combinations of doxycycline and zeocin? Mutations of any type (K, T, G) can arise spontaneously, then establish in the population, and compete with each other depending on two requirements. First, the mutation type must be available (genetic changes causing the phenotype must exist). Second, since we consider large populations, the mutation should be beneficial, improving fitness in the given environment. Despite these intuitive expectations, it is unclear how many mutations of each type will establish in each condition, and how fast.

To address these questions *in silico*, we developed two complementary modeling approaches: a simple mathematical model and a detailed computational simulation framework (see the Computational Models.zip file and the Appendix for detailed descriptions). The two models serve to test the robustness of results to various modeling approaches. The simple model was more general and faster, allowing more extensive parameter scans. On the other hand, the simulation framework allowed testing how specific details of experimental evolution would affect the evolutionary dynamics, and provided more detailed results. We initiated both models with a population of ancestral (wild-type) PF cells, aiming to find out the number and type of mutations that establish and when the ancestral genotype disappears. We modeled 20 days of evolution in each environment indicated by the colored squares in Fig 1B.

The simpler model described population dynamics by a system of ordinary differential equations (ODEs), assuming constant population size and mutation rate. We characterized wild-type and mutant cells by a single parameter: their fitness (exponential growth rate), determined from the fitness landscapes in Fig 1B. For example, we assumed that K mutants had cellular fitness corresponding to null expression in Fig 1B. T-type mutant cells altered their fitness

randomly to a level corresponding to intermediate expression on the cellular fitness landscapes. Finally, G-type mutants increased their fitness randomly, up to a level they would have without zeocin. This simpler model could predict how fast the wild-type genotype disappears from the population. It could also forecast the mutation type (K, T, G) that predominantly replaces the wild type in each condition. However, it could not predict the number of distinct mutant alleles in the evolving population. Moreover, it lacked potentially important experimental details, such as periodic resuspensions and phenotypic switching.

To test the importance of such additional details, the detailed simulation framework captured multiple experimentally relevant aspects of evolution. For example, cells could switch between On and Off states with experimentally inferred rates (Appendix Table S1). K, T, and G mutations with altered switching and growth rates entered the population as single cells at a constant, but adjustable rate $\mu$ per cell per generation (Fig 2A; Appendix Fig S1C). K-type mutants could not switch On, and thus had no cellular fitness costs in doxycycline. T-type mutants switched On at a randomly reduced rate, and thus had diminished cellular fitness costs from PF gene expression. G-type mutant cells had randomly increased drug resistance without any change in switching rates. We simulated periodic resuspensions by repeatedly reducing the cell population size to $10^6$. We considered cells to be initially drug- and inducer-free, and allowed them to gradually take up zeocin and doxycycline. This simulation framework could predict the number of distinct mutant alleles, in addition to the characteristics predicted by the simpler model.

Both models had three free parameters: the rate of potentially beneficial mutations $\mu$, and the input probabilities P(G) and P(T) of a given mutation being of type G or T, respectively. Once known, these parameters also define the probability of a mutation to be of type K: P(K) = 1 – P(G) – P(T). We note the difference between the rate and probability of a mutation: for example, the probability of P(K) could be equal to 1, while its rate $\mu$P(K) is much < 1 per genome per generation. Figure 2A depicts the effect of each mutation type, illustrating the relationships among the free parameters. We extracted the rest of the parameters (Appendix Table S1) from experimental measurements (see the Appendix) and kept them fixed.

Using these models, we studied how the three free parameters affected three features of evolutionary dynamics: the ancestral genotype's half-life, as well as the type and number of mutant alleles in each condition (Fig 2; Appendix Figs S2 and S3). We started by studying the ancestral genotype's half-life in each model, scanning each free parameter systematically (Fig 2B; Appendix Figs S2B and S3B). The models consistently indicated (Fig 2B) that the ancestral genotype disappeared fastest in conditions with steep monotone cellular fitness landscapes (Fig 1B, D0Z2 and D2Z0). In contrast, the ancestral genotype remained in the population longer in peaked cellular fitness landscapes (Fig 1B, D2Z2 and DiZ2). Finally, the majority of cells were still genetically ancestral after 20 days in DiZ0, which has the most gradual cellular fitness landscape (Fig 1B, DiZ0). The time when the ancestral genotype disappeared in various environments depended differently on the mutation probabilities P(K), P(T), P(G) (Appendix Fig S3B). For example, the ancestral genotype disappeared later in D2Z2 when we lowered P(T). Likewise, lowering P(G) prolonged the ancestral genotype's

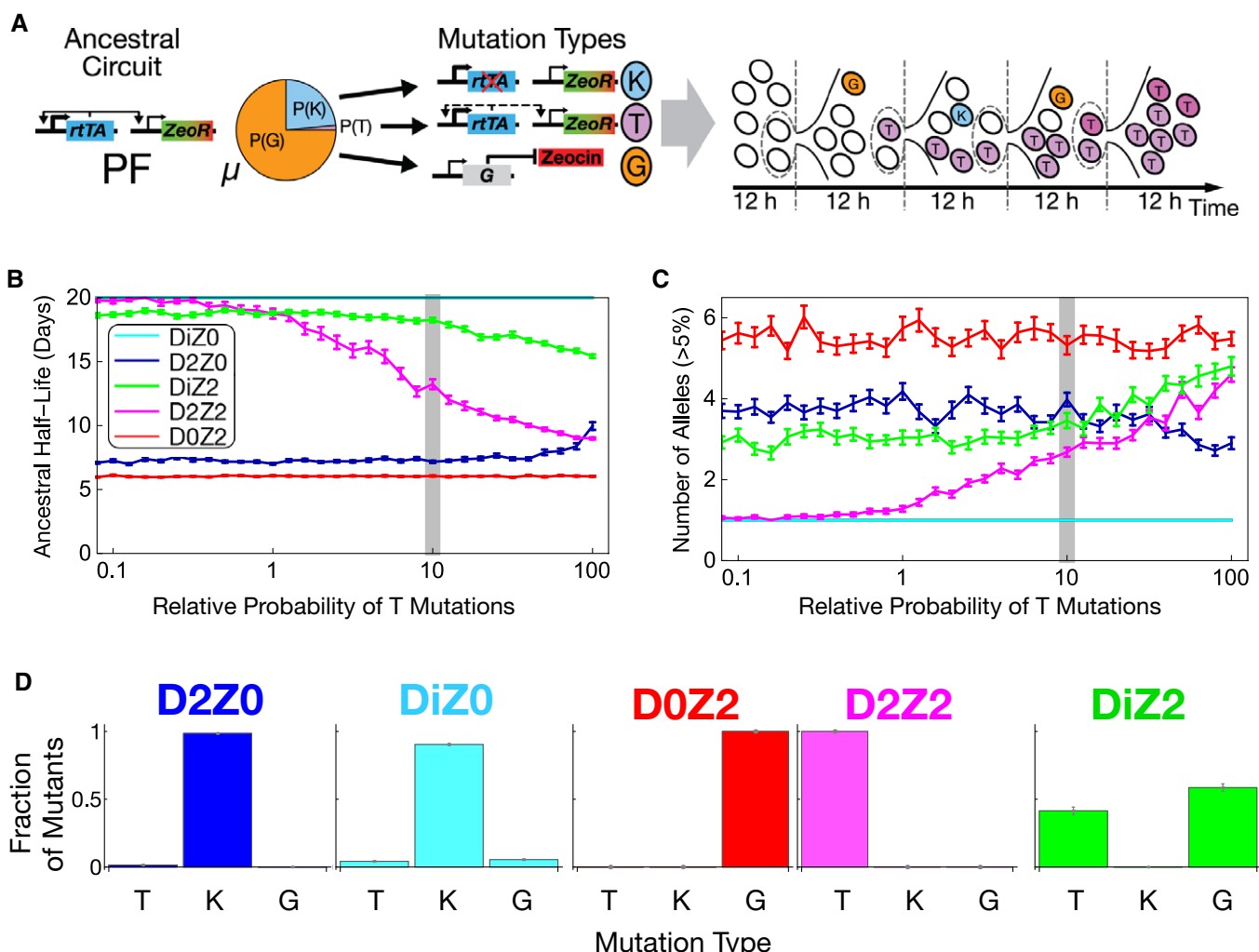

**Figure 2.  Simulation framework predicts evolutionary dynamics.**

A   Simulating the initial steps of evolution. Three types of potentially beneficial mutations (with an overall rate μ) enter the ancestral population of yeast cells that initially carry the intact PF gene circuit. Each cell can divide and mutate, producing new genotypes with altered fitness that can belong to three different types. The first two types are knockout (K) and tweaking (T) mutations. They eliminate rtTA's regulator activity and toxicity completely or partially, respectively. The third type includes extra-rtTA or generic (G) mutations that cause zeocin resistance independently of rtTA. In the models, we consider exponential growth with random elimination of cells or periodic resuspensions to control population size. Empty circles represent intact PF cells, while blue, magenta, and orange circles represent K, T, and G mutants, respectively. These mutations can arise, be lost, or expand in the population.

B   The speed at which mutants take over the population in each simulated condition is measured as the ancestral genotype's half-life (the time until only 50% of the population carries the ancestral genome). $N = 100$; mean $\pm$ SEM in each simulated condition: D2Z0, DiZ0, D0Z2, D2Z2, and DiZ2. In these plots, we fixed $\mu_{-z} = 10^{-6.2}$ or $\mu_{+z} = 10^{-5.4}$/genome/generation (for no zeocin and zeocin, respectively) and P(G) = 0.75. Therefore, P(T) = 0.25 − P(K). On the horizontal axis, we show the probability of T mutations among intra-rtTA mutations: P*(T) = P(T|¬G), which scales P(T) four-fold up such that its maximum is 1 instead of 0.25. The gray bar denotes the value used for time course simulations in subsequent figures. The parameter set for the gray bar on this and the following panels is P(T) = 0.025; P(K) = 0.225; and P(G) = 0.75.

C   Number of established mutations with frequency > 5% at day 20. $N = 100$; mean $\pm$ SEM in each simulated condition: D2Z0, DiZ0, D0Z2, D2Z2, and DiZ2. Parameters, axes, and gray bar: as in (B).

D   Population fractions of T-, K-, and G-type mutations at day 20, for the parameters corresponding to the gray bar, as indicated above.

presence in the populations in D0Z2. These observations confirmed the expectation that the most beneficial mutation in each condition dictates evolutionary dynamics. Overall, we hypothesized based on these results that the ancestral PF gene circuit should disappear fastest in D2Z0 and D0Z2, followed by DiZ2 and D2Z2, and finally in DiZ0. Making these predictions required quantitatively understanding the fitness properties and genetic structure of the PF gene circuit. Without modeling, it would have been impossible to obtain

quantitative estimates of the speeds at which mutants establish and take over the evolving population.

In general, K, T, and G allele frequencies at the end of simulated time courses did not match the input probabilities of P(K), P(T), and P(G) mutations. Rather, each condition favored different mutation types as long as they were available (Fig 2D; Appendix Figs S2 and S3). For example, in D2Z0, nearly all mutations were K-type even if K mutations were unlikely to enter the population. T

mutations established exclusively in D2Z2, while in DiZ2 they appeared alongside G mutations. In DiZ0, K or T mutations established late and spread slowly, with parameter-dependent relative fractions. Finally, only G alleles could establish in D0Z2. To conclude, both models predicted the environment-specific dominance of various mutation types at 20 days, irrespective of the relative supplies of different mutation types. The most likely cause is each condition selecting one mutation type so strongly that the final outcome of evolution (but not its dynamics) becomes quasi-deterministic. The long-term dominance of specific mutants in each condition might have been intuitively inferable from the fitness properties and genetic structure of the PF gene circuit. However, modeling is indispensable to understand the evolutionary dynamics of mutants arising, establishing and competing before reaching the final state.

Finally, we used the simulation framework to determine the number of alleles over 20 days in each condition (Fig 2C). This is perhaps the least intuitive result that could not have been predicted without computation. The simulations indicated that the number of alleles exceeding a certain frequency depended strongly (sometimes non-monotonically) on the overall mutation rate as well as the availability of individual mutations (Appendix Fig S3A). The dependence of allele numbers on simulation parameters should allow parameter estimation once experimental allele data are available.

In summary, based on mathematical and computational models, we hypothesized that the ancestral PF gene circuit should disappear from the population fastest in conditions D2Z0 and D0Z2, followed by D2Z2 and DiZ2, and lastly DiZ0. In addition, we conjectured that K, T, and G mutations should predominate in D2Z0, D2Z2, and D0Z2, respectively, whereas mixtures of T and G genotypes should prevail in DiZ2. Mutations (K or sometimes T) should be slow to establish in DiZ0, causing the ancestral genotype to remain in the majority even at 20 days. To test these hypotheses, we evolved three replicate PF yeast cell populations in five conditions (DiZ0, D2Z0, D0Z2, DiZ2, D2Z2) corresponding to the colored squares on the population fitness landscape in Fig 1B. We also evolved cells in the control condition D0Z0, where we found only one barely detectable, low-frequency synonymous substitution (Appendix Table S2). We observed directly the relationship between gene expression and fitness by daily fluorescence and cell count measurements over the course of these experiments. For various experiments and on multiple days, we collected samples for whole-genome and traditional (Sanger) sequencing to reveal the mutations underlying the observed gene expression changes.

## Scenario (i): reproducible circuit failure from gratuitous circuit response

To test the fate of a new stress defense module that becomes costly by gratuitously responding to an otherwise harmless environmental change, we grew PF yeast cells in inducer doxycycline without antibiotic (D2Z0), resuspending every 12 h. In this condition, fluorescence first rose and then began to decline toward the basal level in < 1 week (~40 generations) for all three replicate populations (Fig 3A). The fluorescence decline continued until gene expression was indistinguishable from that of uninduced cells by the end of the experiment, consistent with the effect of K-type mutations. As fluorescence levels dropped, population growth rate increased significantly (see the Source Data for Fig 3A), indicating that the initial cost of futile response disappeared. These concurrent fluorescence and fitness changes agreed with the leftward hill climb on the blue landscape in Fig 1B (black arrow underneath D2Z0) expected for K-type mutations.

To uncover the genetic mechanism(s) underlying these fluorescence and fitness changes, we combined whole-genome and Sanger sequencing (see the Appendix). Our analysis revealed four competing mutations inside the rtTA coding sequence that jointly accounted for most of replicate population #1 already at Day 9 (Fig 3C and D "12 h-1"; Appendix Table S3), and eliminated the ancestral genotype by the end of the experiment. The same happened in the other two replicate experiments as well (Fig 3C–E "12 h-2,3"; Appendix Table S3). This is consistent with computationally predicted K-type mutations eliminating rtTA toxicity, along with its transcription-activating function. We detected no mutations in other parts of the genome, although we cannot rule out the possibility of mutations in repeat regions or large duplications/deletions that are notoriously difficult to detect by whole-genome sequencing (Appendix Fig S4D and E). We repeated the evolution experiment with 24-h resuspensions and observed similar fluorescence and fitness changes, along with rtTA coding sequence mutations, except that they occurred faster (Fig 3C "24 h-1,2,3"; Appendix Fig S4A–C, Appendix Table S3). Four of these mutations (three STOP codons

**Figure 3. Evolutionary dynamics of PF cells in D2Z0 and DiZ0, corresponding to scenario (i): futile response to harmless signal.**

A    Time-dependent changes in the fluorescence distributions (blue heatmaps), average fluorescence (blue circles), and average, mixed population fitness (blue squares). Data were collected as PF cells evolved in condition D2Z0 (2 μg/ml doxycycline and no zeocin) in three replicate experiments. Average fluorescence and fitness values in control condition D0Z0 are also shown as black crosses for reference. Both the fluorescence ($P$ = 0.00019) and fitness ($P$ = 0.003959) were significantly different in populations evolving in D2Z0 compared at Days 4 and 21 ($t$-test, see the Materials and Methods).

B    The same measurements as in (A), but for PF cells evolving in condition DiZ0 (0.2 μg/ml doxycycline and no zeocin, cyan heatmaps) in three replicate experiments. The fluorescence ($P$ = 0.0144526) was significantly different, but the fitness ($P$ = 0.2459) was not in populations evolving in DiZ0 when compared at days 4 and 21. Pairwise comparisons with the same days in D0Z0 showed no significant fitness differences (see the source data).

C    Intra-rtTA mutations observed in conditions D2Z0 (blue lines) and DiZ0 (light blue lines) mapped along the rtTA activator within the PF gene circuit sequence. The five lines of annotation indicate the following: (i) basepair coordinates relative to the rtTA translation start site (+1); (ii) nucleotide substitution; (iii) amino acid substitution; (iv) in which experiment the allele was found; and (v) allele fractions at Day 19 inferred from sequencing. If there was a deletion or duplication, the first two lines represent its range. *: STOP codon; Δ: deletion; Dupl: duplication. No extra-rtTA mutations were identified in these conditions. Clones selected for phenotyping are underlined and numbered in blue.

D, E    Time-dependent allele frequencies for mutations observed in conditions D2Z0 (D), and DiZ0 (E), replicate experiment #1. The way we used sequencing data to draw allele frequencies and the lines connecting them is explained in the Mutation time course reconstruction section of the Materials and Methods.

F, G    Time-dependent allele frequencies from simulations using mutation parameter values reflecting experimental conditions.

Source data are available online for this figure.

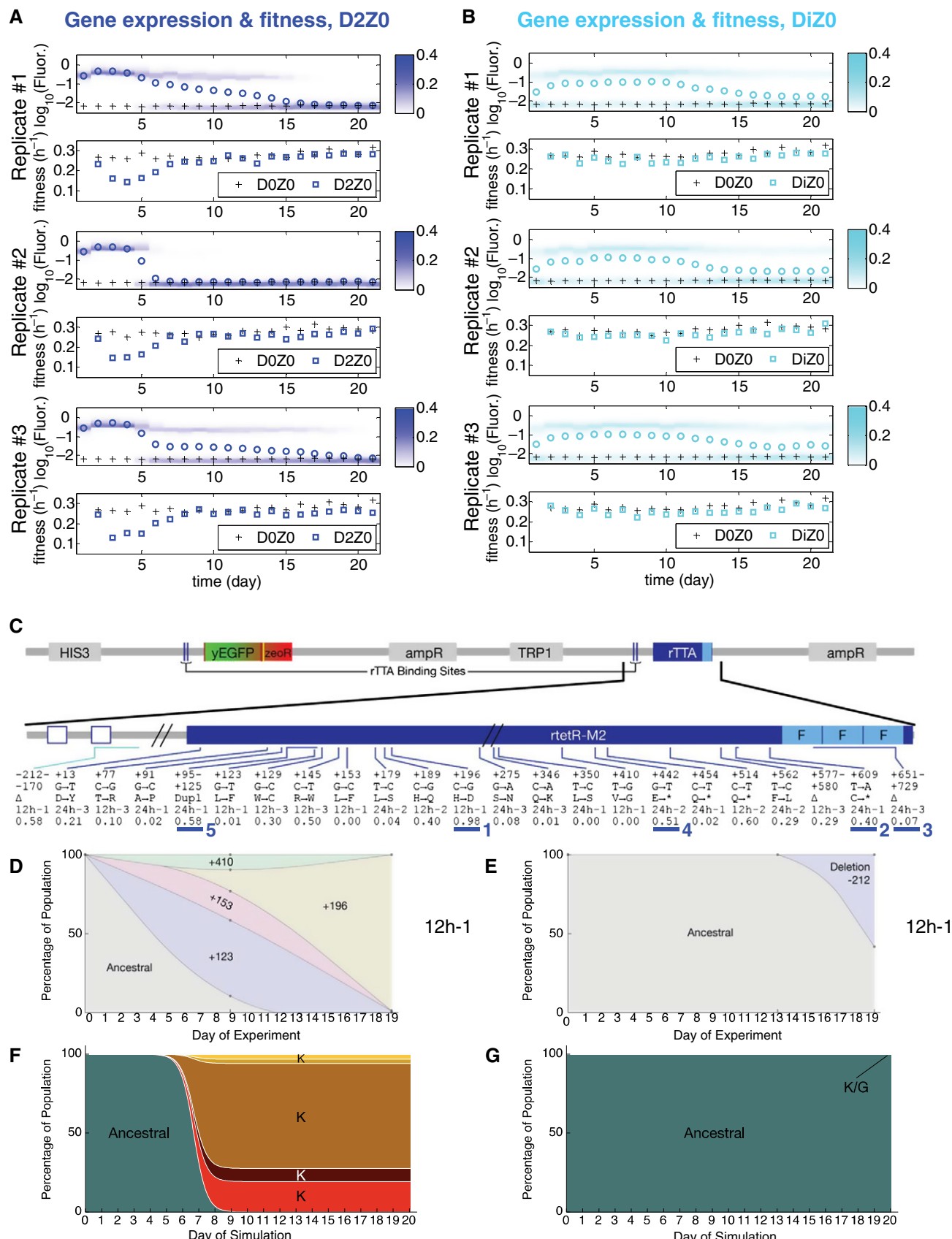

**Figure 3.**

and a 78-base pair deletion) truncated and eliminated all three activator domains of rtTA, further supporting the K-type loss of rtTA function.

Decreasing the inducer (doxycycline) concentration from 2 to 0.2 µg/ml should diminish rtTA toxicity. Selection in this condition should be weaker (Fig 1B), lowering the chances of beneficial mutations establishing in DiZ0 compared to D2Z0. To test these predictions, we evolved three cultures in the DiZ0 condition (Fig 3B). In agreement with computational predictions (Fig 2A and B; Appendix Figs S2 and S3), the fraction of On cells started declining slowly only toward the end of the experiment. This resulted in a statistically significant change in fluorescence, but not in fitness. Moreover, Sanger sequencing at the end of the experiment revealed a single intra-circuit deletion at 58% frequency (Fig 3C), which eliminated one of the two *tetO2* operator sites upstream of rtTA. This suggests a T-type mutation (since one *tetO2* site remained intact) targeting a regulatory region rather than protein-coding sequence. We detected no mutations elsewhere in the genome.

In summary, these experimental observations confirmed the computational predictions that a steep, monotonically decreasing cellular fitness landscape (Fig 1B, D2Z0, blue shading; Fig 3F and G) reproducibly selects for lower gene expression. The effect of these mutations is to decrease gene expression unidirectionally by either eliminating or reducing the fraction of On cells. Thus, deleterious network activation favors mutations that prevent or reduce switching into the slow-growing On state. We selected five individual genotypes (underlined with blue in Fig 3C) for testing whether their gene expression and fitness are consistent with K-type mutations (see below the section on phenotyping).

## Scenario (ii): gaining gene expression for an initially unresponsive gene circuit

To test what happens if a stress defense module cannot induce when needed during harmful stress, we grew cell populations in 2 mg/ml zeocin (D0Z2). The lack of inducer in this condition forced all cells to be in the drug-sensitive Off state. Consequently, the tradeoff between elevated expression and drug resistance specific to the PF gene circuit was absent in D0Z2. Early in the course of evolution, we observed a substantial, statistically significant drop in population fitness compared to untreated cells (Fig 4A), indicating the gene circuit's inability to respond to stress. Yet, some cells must have had enough drug resistance to survive, because the growth rates of cultures started to recover after ∼4 days (Fig 4A). At the same time, yEGFP::ZeoR expression increased significantly compared to control cultures maintained in D0Z0 (Fig 4A). This difference remained statistically significant even after correction for multiple comparisons, particularly toward the end of the experiment. We observed similar trends with 24-h resuspensions (Appendix Fig S5). Thus, the evolving cell population moved repeatedly upward in gene expression and drug resistance space, toward the cellular fitness maximum in Fig 1B (black arrow underneath D0Z2). In contrast, cells lacking the *zeoR* gene never recovered in the same level of zeocin, while cells with higher basal yEGFP::ZeoR expression recovered faster (Appendix Fig S5C–E).

Next, we sought mutations explaining the observed fluorescence and fitness changes. In sharp contrast with D2Z0, we found no

mutations in either rtTA or its regulatory region. Instead, we detected two extra-rtTA, but intra-circuit mutations overall in six replicate experiments (Fig 4D; Appendix Table S4), one of which eliminated a *tetO2* operator site upstream from *yEGFP::zeoR*, while the other was a synonymous substitution in an arginine codon within the *zeoR* coding region. Additionally, sequencing revealed multiple extra-circuit mutations (Fig 4B) and linkage between the intra-circuit *tetO2* deletion and some extra-circuit alleles (Fig 4B and D). This raised the possibility that intra- and extra-circuit mutations jointly detoxify the cells in a manner consistent with G-type mutations. The real number of extra-circuit mutations could be higher, considering the difficulty of detecting certain mutation types by high-throughput sequencing. In addition, some adaptation in D0Z2 could also have occurred through native stress responses or non-genetic selection of the high-expressing tail of the basal yEGFP::ZeoR distribution.

Altogether, these data suggest that as long as cells with a potentially beneficial, but inoperative module have some basal resistance to survive, they can later activate the module and acquire drug resistance by genetic mutations (Charlebois *et al*, 2011). Apparently, this happens through mutations both inside and outside of the module, genetically integrating it with the host. This effect seems dependent on the presence of *zeoR*, since cells lacking this gene do not survive in D0Z2 (Appendix Fig S5C and D). An interesting question is whether drug resistance gained from these mutations involves some cost. To answer this question and test whether the mutants are indeed G-type, we selected and characterized six individual genotypes underlined with red in Fig 4D (see the section on phenotyping below).

## Scenario (iii): optimization of gene expression under opposing evolutionary pressures

To test what happens when a module responds to stress non-optimally, we exposed the cells to both inducer and antibiotic. In these conditions, there is a cellular fitness peak at intermediate gene expression (Fig 1B, DiZ2 and D2Z2, green and magenta shading), in contrast to the monotone cellular fitness landscapes in conditions with only inducer (D2Z0 and DiZ0) or only antibiotic (D0Z2). The cellular fitness peak indicates opposing selection pressures from zeocin toxicity and the fitness cost of rtTA expression: zeocin selects for increased gene expression, while rtTA toxicity selects for diminished rtTA function and thus decreased gene expression (Fig 1B, arrows underneath DiZ2 and D2Z2). These selection pressures act on two cell subpopulations flanking a cellular fitness peak (Fig 1B, D2Z2). Therefore, fitness improvement in DiZ2 and D2Z2 requires adaptation toward an intermediate "sweet spot" of expression. K mutations cannot achieve this since they completely disrupt rtTA function.

In D2Z2, average fluorescence decreased while fitness increased significantly for all replicate cultures (Fig 5A), albeit by a lesser extent and more slowly than in D2Z0 (Fig 3A), as predicted computationally. Sequencing has uncovered only two competing alleles from one replicate culture, each affecting a distinct PF gene circuit component. Sequencing samples from the other two replicate experiments then revealed D2Z2-specific mutations that repeatedly occurred in the same rtTA loci: the 5′ untranslated rtTA region and the 225th basepair of rtTA (Fig 5C; Appendix Fig S6A and B;

                    

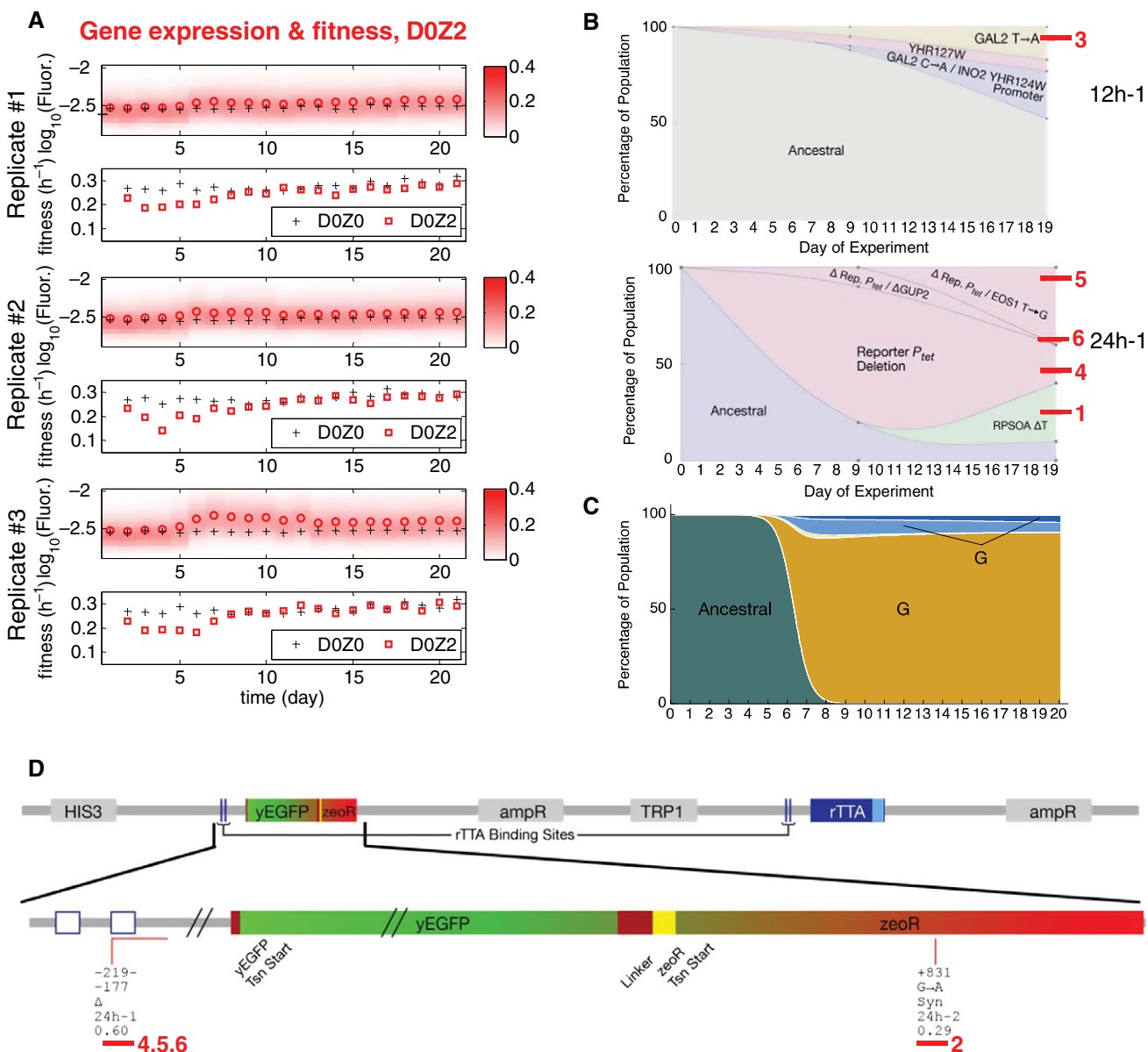

**Figure 4. Evolutionary dynamics of PF cells in D0Z2, corresponding to scenario (ii): lack of response when needed.**

A  Time-dependent changes in the fluorescence distributions (red heatmaps), average fluorescence (red circles), and average, mixed population fitness (red squares) as PF cells evolve in condition D0Z2 (no doxycycline and 2 mg/ml zeocin) in three replicate experiments. Black crosses, as in Fig 3. Both the fluorescence ($P = 0.014323$) and fitness ($P = 0.002244$) were significantly different in populations evolving in D0Z2 when compared at days 4 and 21 (dependent samples *t*-test, see the Materials and Methods). In addition, at many time points, the fluorescence difference from the ancestral PF was statistically significant (independent samples *t*-test, see the Materials and Methods). These statistical differences persisted even after correcting for multiple comparisons. The same was true for fitness at early time points (up to Day 6).

B  Time-dependent allele frequencies for mutations observed in condition D0Z2, replicate #1. Top: whole-genome sequencing from a 12-h resuspension experiment. Bottom: whole-genome sequencing combined with Sanger sequencing of clonal isolates from the same 24 h resuspension experiment, indicating linkage between intra- and extra-PF mutations. Among the observed mutations, *INO2* is a regulator of phospholipid biosynthesis that lowers stress resistance, and *YHR127W* function is unknown, but is synthetic lethal with *ARP1*, which mediates resistance to multiple stresses. Red bars and numbers indicate clones selected for phenotyping. The way we used sequencing data to draw allele frequencies and the lines connecting is explained in the Mutation time course reconstruction section of the Materials and Methods.

C  Time-dependent allele frequencies from simulations using mutation parameter values reflecting experimental observations.

D  Extra-rtTA, but intra-circuit mutations observed in condition D0Z2 (red lines) mapped along *yEGFP::zeoR* within the PF gene circuit sequence. The five lines of annotation indicate the following: (i) basepair coordinates relative to the yEGFP::zeoR translation start site (+1); (ii) nucleotide substitution; (iii) amino acid substitution; (iv) which experiment the allele was found; and (v) allele fractions at Day 19 inferred from sequencing. If there was a deletion, the first two lines represent its range. Δ, deletion; Syn, synonymous. Two extra-rtTA, but intra-circuit mutations were identified in this condition. Clones selected for phenotyping are underlined and numbered in red.

Source data are available online for this figure.

Appendix Table S5). Another mutation truncated rtTA by a STOP codon in the last activator domain but left the two other domains intact, suggesting a T mutation with diminished rtTA function and toxicity, while still maintaining a zeocin-resistant, yEGFP::ZeoR-expressing subpopulation.

Next, we studied how lower but nonzero rtTA toxicity affects evolutionary dynamics for a peaked fitness landscape, propagating PF cells in 0.2 μg/ml doxycycline and 2 mg/ml zeocin (Fig 1B, DiZ2). The addition of zeocin selects against low-expressing Off cells, reshaping the bimodal distribution seen in DiZ0, so that the fraction of On cells increases in DiZ2 (compare black histograms overlaid with cyan and green shading, DiZ0 and DiZ2 in Fig 1B). These high expressors thus survive in stress and can maintain the population until more potent drug resistance mutations arise (Charlebois *et al*, 2011). Indeed, fitness decreased only slightly during evolution in DiZ2 (Fig 5B). After Day 7, fluorescence seemed to decrease slowly while fitness crept up throughout the time course. These changes were not statistically significant when we compared fitness and fluorescence values at Day 4 and Day 21 along the DiZ2 time course. However, we found that the fitness in DiZ2 was significantly lower than in D0Z0 at several time points, which remained true even after correcting for multiple comparisons.

DiZ2 was the only condition where mutations affecting both rtTA and extra-circuit loci established (Fig 5E). The intra-circuit mutation was a *tetO2* site deletion from the rtTA promoter, eliminating the other *tetO2* site compared to the deletion in DiZ0. Additionally, we detected three extra-rtTA mutations, one of them linked to the *tetO2* deletion. In general, these findings indicated that peaked fitness landscapes selected for T-type mutations, while also allowing for G-type mutations, as predicted computationally (Figs 2D and 5F and D). We confirmed these mutation types by testing whether the mutations weakened rtTA activity, without eliminating it (see below).

**Phenotyping reveals fitness-improving network characteristics**

In contrast to the D0Z0 control condition where fitness and gene expression changes were statistically non-significant (Appendix Fig S6D and E), these quantities changed significantly in other conditions tested (Figs 3, 4 and 5). These changes generally involved mixed populations composed of different genotypes competing with each other. To characterize individual genotypes in isolation, we measured gene expression levels and population fitness of clonal isolates from the last day of the evolution experiments.

First, we studied five clonal isolates from the last day of the D2Z0 time course (underlined in blue in Fig 3C), to test whether they carry K-type mutations. If this is true, then they should be uninducible and their fitness should not depend on doxycycline. Therefore, we quantitatively characterized the effect of doxycycline on the fluorescence and fitness of these clones. Thus, we defined the fitness effect of doxycycline as $\log_{10}$[(fitness with doxycycline)/(fitness without doxycycline)]. Likewise, we defined the effect of doxycycline on fluorescence as $\log_{10}$[(fluorescence with doxycycline)/(fluorescence without doxycycline)]. Based on these measures, we found that all five clones isolated from the D2Z0 inducer-only condition were fitter (Fig 6A, top panel) than the PF ancestor and were uninducible (Fig 6A, middle panel). These properties matched the characteristics of K-type mutations predicted computationally to dominate in D2Z0 (Fig 6A). Sanger sequencing of clonal isolates from the middle and the end of the D2Z0 evolution time course indicated that each K-type mutation occurred individually, without linkage to other mutations. Some of these clones were also fitter in D0Z0 compared to the ancestral strain, suggesting additional adaptation to growth in minimal medium (Lenski & Travisano, 1994; New *et al*, 2014) after eliminating the rtTA toxicity.

Next, we studied clonal isolates from the last day of the D0Z2 time course (Fig 4B and D) to test whether they are zeocin-resistant. We quantitatively characterized the effect of zeocin on the fitness of these clones as $\log_{10}$[(fitness with zeocin)/(fitness without zeocin)]. To determine whether zeocin resistance arose from higher yEGFP::zeoR expression, we also defined the gene expression increase in these clones as $\log_{10}$[(fluorescence of evolved clone in D0Z0)/(fluorescence of PF ancestor in D0Z0)]. We found that all clones isolated from the zeocin-only condition (underlined in red in Fig 4B and D) had higher fitness in zeocin (D0Z2) compared to ancestral PF cells (Fig 6B, top panel). The cause of zeocin resistance was higher *yEGFP::zeoR* gene expression even in the condition D0Z0, without zeocin (Fig 6B, middle panel and Fig 6E). These observations are consistent with G-type mutations, predicted computationally to dominate in D0Z2. *yEGFP::zeoR* gene expression in all clones shifted significantly upward, obeying the selection pressure (Fig 1B, black arrow underneath D0Z2). Some clones had two linked mutations, one of which was within the PF gene circuit, while the other was outside of it. We found no mutations for one zeocin-resistant clone

**Figure 5. Evolutionary dynamics of PF cells in D2Z2 and DiZ2, corresponding to scenario (iii): suboptimal response.**

A     Time-dependent fluorescence distributions (magenta heatmaps), average fluorescence (magenta circles), and mixed population fitness (magenta squares) as PF cells evolve in condition D2Z2 in three replicate experiments. Black crosses, same as in Fig 3. Both the fluorescence ($P = 0.0003157$) and fitness ($P = 0.010568$) were significantly different in populations evolving in D2Z2 when compared at days 4 and 21. Statistical test: as above.

B     The same measurements as in panel (A), but for PF cells evolving in condition DiZ2 in three replicate experiments. Neither fluorescence ($P = 0.95$), nor fitness ($P = 0.087$) was significantly different in populations evolving in DiZ2 when compared at days 4 and 21. Pairwise comparisons with the same days in D0Z0 showed significant fitness differences, many of which remained true even after correction for multiple comparisons.

C     Intra-circuit mutations observed in conditions D2Z2 (magenta lines) and DiZ2 (green lines) mapped along the rtTA activator within the PF gene circuit sequence. The five lines of annotation indicate: (i) basepair coordinates relative to the rtTA translation start site (+1); (ii) nucleotide substitution; (iii) amino acid substitution; (iv) which experiment the allele was found; and (v) allele fractions at Day 19 inferred from sequencing. If there was a deletion, the first two lines represent its range. *, STOP codon; Δ, deletion. While no extra-rtTA mutations were identified in condition D2Z2, a few were found in DiZ2 (see E). Clones selected for phenotyping are underlined and numbered.

D, E  Time-dependent allele frequencies for mutations observed in conditions D2Z2 (D) and DiZ2 (E), replicate #1. The way we used sequencing data to draw allele frequencies and the lines connecting is explained in the Mutation time course reconstruction section of the Materials and Methods.

F, G  Time-dependent allele frequencies from simulations using mutation parameter values reflecting experimental conditions.

Source data are available online for this figure.

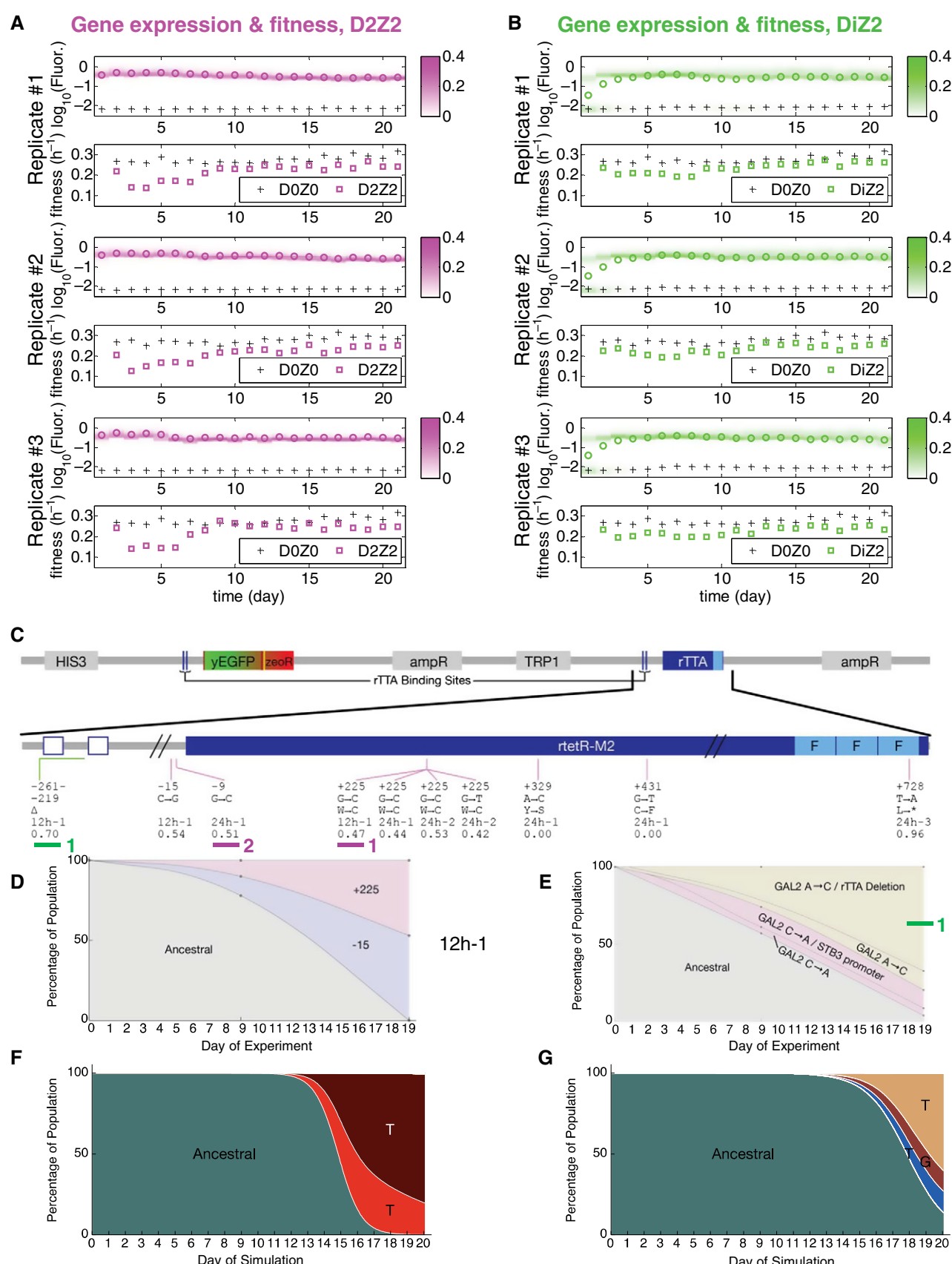

**Figure 5.**

in any locus tested by Sanger sequencing, suggesting extra-circuit mutation(s) undetectable by either whole-genome or targeted Sanger sequencing. These results indicate that adaptation in D0Z2 recurrently involves mutations causing PF gene expression increase. This is surprising considering that mutations could have just upregulated native stress resistance pathways without involving the PF gene circuit.

Considering the original tradeoff between the cost of gene expression and benefit of drug resistance in the PF gene circuit, we asked whether a similar tradeoff may apply to drug-resistant genotypes evolved in D0Z2. Interestingly, adaptation by elevated basal yEGFP::zeoR expression tended to cause a fitness cost in D0Z0, when zeocin was absent (Fig 6B, bottom panel). The sources of these new fitness costs are unclear, but they are not rtTA-related because doxycycline was absent. This suggests that a novel tradeoff appeared between evolved stress resistance and growth in the absence of stress (Fig 6G). This new tradeoff is reminiscent of the original tradeoff in the ancestral PF gene circuit, where higher expression was also costly, but protective in the presence of antibiotic.

We similarly characterized the effects of doxycycline and zeocin on the fitness and gene expression of two clones isolated from D2Z2 experiments (underlined in magenta, Fig 5C). We found that both clones isolated from D2Z2 had reduced inducer sensitivities (Fig 6C), requiring higher doxycycline than the PF ancestor to reach a given gene expression level (Fig 6C, middle panel). Generally, the gene expression distributions of these clones were enriched in Off cells (Appendix Fig S7C and D). These changes were associated with lower doxycycline toxicity (Fig 6C, top panel, blue bars), while the cells still maintained drug resistance in doxycycline (Fig 6C, bottom panel, magenta bars). These characteristics were consistent with T-type mutations, as predicted computationally to dominate for peaked cellular fitness landscapes. Interestingly, in addition to the increase in Off cells, the On state moved to lower expression, toward the cellular fitness peak in the gene expression space (Appendix Fig S7C and D). We could still fully induce these clones by applying excessive (6 μg/ml) doxycycline levels with zeocin (Appendix Fig S7C and D). Interestingly, all cells were in the On state (fully induced) throughout the 20 days of evolution in D2Z2, but clones

placed in D2Z2 were not. Taken together, these observations suggest that gene circuit bistability may have trapped mutant cells in the On state during evolution if the mutation arose in cells that were On.

Finally, we studied the single clone isolated from the DiZ2 experiments (*tetO2* deletion; underlined in green, Fig 5C). These cells required only slightly higher doxycycline levels for induction than the PF ancestor. Yet, once induced, they rose to higher mean expression level than the ancestor (Fig 6D; Appendix Fig S7A). Moreover, the two peaks in the bimodal gene expression histograms approached each other for this clone (Fig 6F; Appendix Fig S7A), both shifting toward the cellular fitness peak in Fig 1B, DiZ2 as dictated by selection. This is a unique example of noisy gene expression evolving under opposing selection pressures (Fig 1B, black arrows underneath DiZ2). Essentially, although evolution altered the gene expression, its distribution still remained bimodal, with a similar mean. While this mutation apparently alters rtTA function, it is different from the T-type mutations assumed in computational models (which did not account for shifting of peaks). This unique type of adaptation has no equivalent in phenotypically homogeneous populations with unimodal gene expression distributions.

To measure the phenotypic effects of the observed mutations in isolation from potential changes in the genetic background, we reconstructed the mutations $rtTA_{+225G \to C}$ (D2Z2 clone #1) and $rtTA_{-9G \to C}$ (D2Z2 clone #2) in the ancestral PF background (Appendix Fig S7C and D). The $rtTA_{+225G \to C}$ mutation was slightly inducible in the ancestral background, with a small high-expressing subpopulation at 2 μg/ml doxycycline. Moreover, we could reinduce this clone to nearly full expression using excessive doxycycline concentrations (6 μg/ml) in the presence of zeocin, suggesting that the reconstructed mutation $rtTA_{+225G \to C}$ lowered the dynamic range and sensitivity similar to the clonal isolate. Interestingly, however, the reconstructed $rtTA_{-9G \to C}$ mutation failed to induce even with excessive doxycyline concentrations, suggesting linkage and potential epistasis with some undetectable genetic extra-circuit mutation(s).

Overall, phenotyping validated the prevalence of K, T, G mutation types in different environments, as predicted computationally. Our observations also underscore the potential importance of noise-reshaping T-type mutations in artificial and natural evolution.

---

**Figure 6. Gene expression and fitness characteristics of clonal isolates from various evolved populations.**

A  Phenotype of clones evolved in inducer doxycycline alone (D2Z0, "futile response"). The first bar ("Anc.") corresponds to the ancestral PF cells, and the other bars correspond to clonal isolates from the last time point of the D2Z0 experiment. Top panel: $\log_{10}$-ratio of fitness with doxycycline (D2Z0) relative to no doxycycline (D0Z0). Middle panel: $\log_{10}$-ratio of average fluorescence intensity with doxycycline (D2Z0) relative to no doxycycline (D0Z0). Bottom panel: $\log_{10}$-ratio of average population fitness of each evolved clone relative to the ancestor in no doxycycline (D0Z0). Error bars represent standard deviations around the mean. Stars denote significance at $P < 0.05$ (two-sided $t$-test).

B  Phenotype of clones evolved in antibiotic zeocin alone (D0Z2, "lack of response when needed"). The first bar ("Anc.") corresponds to the ancestral PF cells, and the other bars correspond to mutants. Top panel: $\log_{10}$-ratio of fitness with zeocin (D0Z2) relative to no zeocin (D0Z0). Middle panel: $\log_{10}$-ratio of average fluorescence intensity of each evolved clone relative to the ancestor in no zeocin (D0Z0). Bottom panel: $\log_{10}$-ratio of average population fitness of each evolved clone relative to the ancestor in no zeocin (D0Z0). Error bars and stars as in (A).

C  Phenotypes of two clones evolved in doxycycline and antibiotic zeocin (D2Z2, "suboptimal response"). The bars marked "A." correspond to the ancestral PF cells, and the other bars correspond to mutants. Top panel: $\log_{10}$-ratio of fitness with doxycycline (D2Zy) relative to no doxycycline (D0Zy) either with or without zeocin ($y = 0$ or $y = 2$). Middle panel: $\log_{10}$-ratio of average fluorescence intensity with doxycycline (D2Zy) relative to no doxycycline (D0Zy). Bottom panel: $\log_{10}$-ratio of fitness with zeocin (DxZ2) relative to no zeocin (DxZ0), either with or without doxycycline ($x = 0$ or $x = 2$). Error bars and stars as in (A).

D  Phenotype of the single clone isolated from intermediate doxycycline and antibiotic zeocin (DiZ2, "suboptimal response"). The bars marked "A." correspond to the ancestral PF cells, and the other bars correspond to the mutant clone. Top panel: $\log_{10}$-ratio of fitness with doxycycline (D2Zy) relative to no doxycycline (D0Zy) either with or without zeocin ($y = 0$ or $y = 2$). Middle panel: $\log_{10}$-ratio of average fluorescence intensity with doxycycline (D2Zy) relative to no doxycycline (D0Zy). Bottom panel: $\log_{10}$-ratio of fitness with zeocin (DxZ2) relative to no zeocin (DxZ0), either with or without doxycycline ($x = 0$ or $x = 2$). Error bars and stars as in (A).

E  Gene expression histograms measured in D0Z0 for Clones #4 and #7 (evolved in D0Z2) compared to the PF ancestor (shaded histogram).

F  Gene expression histograms measured in DiZ2 for Clone #1 (evolved in DiZ2) compared to the PF ancestor (shaded histogram).

G  Tradeoff between yEGFP::zeoR expression and zeocin resistance for clones evolved in D0Z2 (red) and DiZ2 (green).

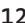

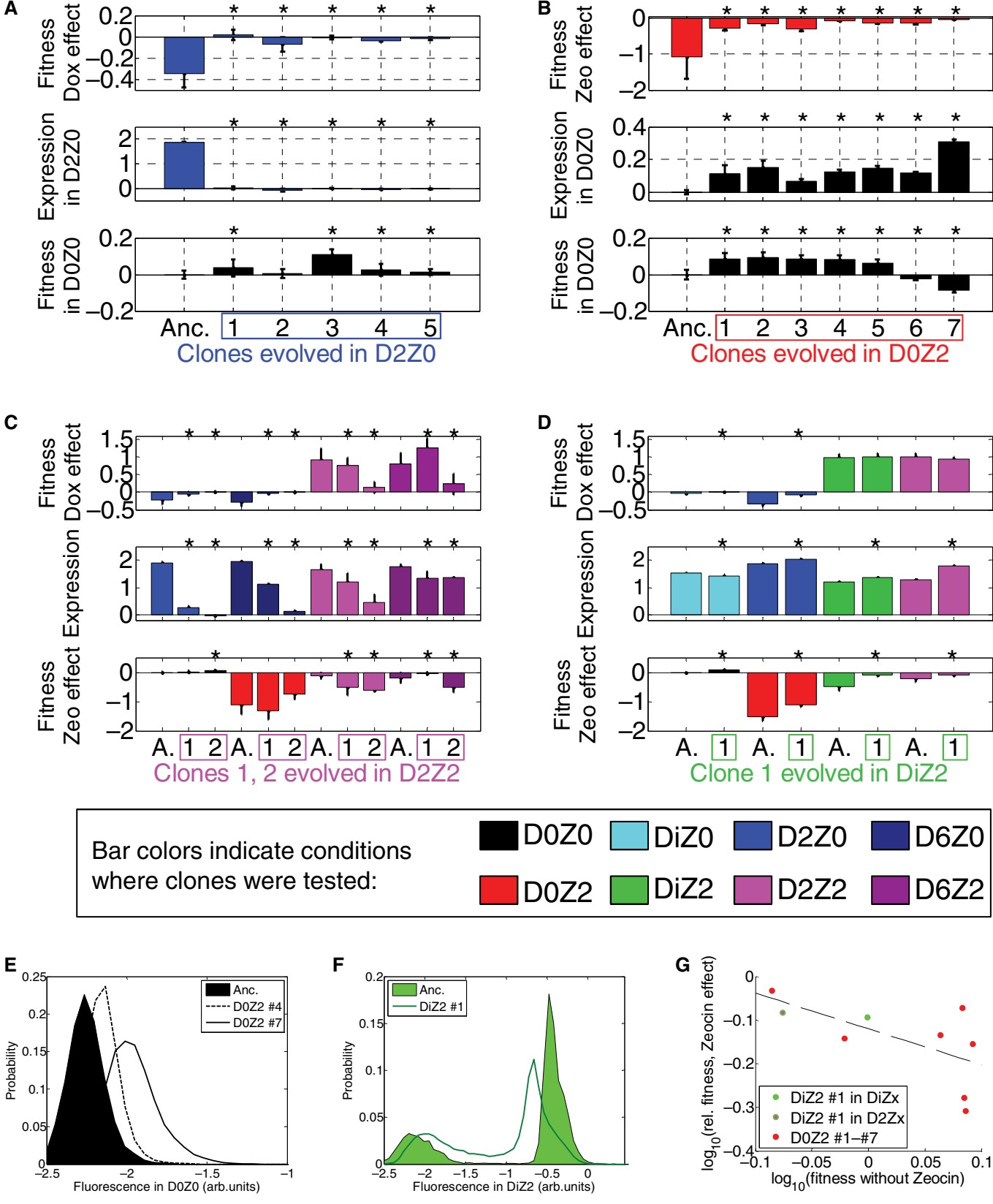

**Figure 6.**

### Additional insights into PF evolutionary dynamics

Experimental evolution and phenotyping validated the major mutation types predicted computationally for each condition. Therefore, we asked whether the computational framework could provide any additional insights into evolutionary forces and mechanisms based on the experimental data.

First, we tried to estimate the rate μ of potentially beneficial mutations using its predicted effect on various allele numbers in several conditions. Interestingly, we could not capture the experimental number of alleles and the half-life of ancestral genotype when we applied the same mutation rate in all conditions. Instead, comparing the results of computational simulations (Fig 4C; Appendix Fig S2D and F) with experimental data suggested slightly higher mutation rate in zeocin than without it (Fig 2). Specifically, the rate of potentially beneficial mutations that matched the data best was $\mu_{-Z} = 10^{-6.2}$ with zeocin compared to $\mu_{+Z} = 10^{-5.4}$/genome/generation without zeocin. This increase is reasonable because zeocin is a DNA-damaging agent that may elevate mutation rates. These beneficial mutation rates are comparable with a recent estimate in yeast of ~$10^{-6}$/genome/generation (Levy *et al*, 2015).

Second, we asked whether we could extract any information about the mutation probabilities P(K), P(T), and P(G). We compared simulation results with experimental data in D2Z0 and D2Z2 conditions where K- and T-type mutations should be prevalent, respectively. Selection for various mutation types is environment-dependent, implying that the number and type of established mutations must depend on the K/T bias in mutations entering the population. For example, while mainly T-type mutations can establish and the K-type is deleterious in D2Z2 (since it forces cells into the drug-sensitive Off state), the opposite is true in D2Z0. Comparing experimentally observed allele numbers with simulation results indicated that incoming T mutations should be in the minority (gray bars, Fig 2B and C) compared to the ~10 times more available K-type mutations. This suggests that only a few, specific rtTA loci can harbor T-type mutations, explaining recurrence of certain mutations in D2Z2. These recurrent mutations must have the rare capability of tweaking protein function and toxicity while still maintaining drug resistance, as predicted computationally and validated experimentally (Fig 6).

Finally, we asked how mutations that arose prior to setting the environmental conditions may have contributed to the outcome of evolution experiments. This was important because the PF cells grew for 24 h in D0Z0 before initiating our evolution experiments. To address this question, we used a variant of the simulation framework that allowed neutral mutations to accumulate for 24 h of growth at the mutation rate $\mu_{-Z} = 10^{-6.2}$/genome/generation. Afterward, we changed the simulated condition to DiZ0, D2Z0, DiZ2, D2Z2, or D0Z2 using values of the free parameters estimated from experimental data. We then computed the contribution of these "preexisting" mutations to the final allele frequencies (Appendix Fig S8). The results indicated that preexisting mutations do not comprise a large fraction of mutant alleles in conditions DiZ0, D2Z2, and DiZ2. On the other hand, in steep monotonic cellular fitness landscapes (D2Z0 and D0Z2), preexisting alleles could comprise approximately 35% of mutant alleles in D2Z0 and ~50% in D0Z2. Nevertheless, the same mutation types dominated in specific conditions with or without pre-existing

mutations. Likewise, the pre-existing mutations did not substantially alter the ancestral genotype's half-life in any condition (Appendix Fig S8).

## Discussion

Stress response networks play key roles in the emergence of drug resistance, from pathogenic microbes to cancer. Typically, stress response incorporates a tradeoff: cells that activate it grow slower in the absence of stress. Therefore, optimality of these networks depends on maintaining the balance between environmental stress and internal response. Yet, it is unknown how quickly, how reproducibly, and through what types of mutations stress response networks evolve to balance the costs and benefits of their response to external stress. What aspects of network evolution are predictable *a priori* and what is required for making predictions is unclear. To address these questions, we studied evolving yeast cells endowed with a synthetic stress response gene circuit that allowed for separate control of the stress and the response by adjusting antibiotic and inducer concentrations, respectively.

Using quantitative knowledge of the PF gene circuit, we developed two computational models to predict specific aspects of evolutionary dynamics in six different environmental conditions. The predicted aspects included the speed at which the ancestral genotype disappears from the population, as well as the types and numbers of mutant alleles that establish in each environmental condition. We validated these predictions by experimental evolution. The agreement between our predictions and experimental findings suggests that cellular and population fitness landscapes can be useful to predict short-term evolution. Critically, our predictive models were based on quantitative knowledge of the fitness and gene expression properties, as well as the genetic structure (design) of the PF gene circuit. Without such knowledge, it would have been impossible to predict what type of mutations arise and how fast. Once this knowledge is acquired, however, cellular and population fitness landscapes (Fig 1B) can be constructed, which are informative for predicting evolutionary outcomes.

We found a connection between the rates at which various potentially beneficial mutations entered the populations and the computationally predicted features of evolutionary dynamics, especially the number of mutant alleles (Appendix Fig S3). This allowed a rough estimation of the relative probabilities of two mutation types to occur spontaneously. We found that mutations eliminating protein function were much more common than mutations fine-tuning protein function (at least for rtTA in these experiments). The availability of various beneficial mutation types depends on DNA sequence and is rarely known *a priori*. We suggest nonetheless that the availability of mutation types could be estimated by comparing computational predictions with actual observations in similar laboratory evolution experiments.

A unifying theme for all environmental conditions was the tradeoff between stress resistance and stress-free growth: genotypes that resisted zeocin tended to grow slower in its absence. Such tradeoffs were inherent by design to the ancestral PF synthetic gene circuit (Fig 1B). However, in D0Z2, yeast adapted using extra-PF mutations that were not subject to the original tradeoff. Most surprisingly, these extra-circuit changes were subject to a different tradeoff,

which resembled the original one in the PF gene circuit (Fig 6G). Essentially, there was a cost for higher yEGFP::zeoR expression, even if caused by extra-circuit mutations. Thus, without the built-in tradeoff within the PF gene circuit, another tradeoff appears through mutations outside of the PF gene circuit. This suggests a fundamental conflict between two different tasks (resistance to stress and fast growth in stress-free conditions), typically resolved by Pareto optimization (Shoval *et al*, 2012). Such "multi-layered" tradeoffs (when multiple ways of coping with stress exist, but each has its own type of tradeoff) may occur frequently in many natural systems, including more complex genetic circuits in other organisms.

The ultimate success of synthetic biology will depend on the long-term practical applicability of synthetic constructs. Despite the growing number of synthetic constructs, their evolutionary stability only recently began to be investigated in *Escherichia coli* (Yokobayashi *et al*, 2002; Sleight *et al*, 2010; Wu *et al*, 2014). As far as we know, this question has not been addressed in eukaryotes. Our work fills this gap and generates insights for building evolutionarily robust eukaryotic gene circuits. The PF gene circuit is based on the rtTA activator, which is widely utilized in eukaryotic synthetic biology. An important insight that we gained was that eukaryotic activators like rtTA are not ideal if gene circuit stability is a concern. There is evidence that eukaryotic activators are generally toxic (Baron *et al*, 1997), which seems to be true for some prokaryotic components as well (Tan *et al*, 2009). To address this problem, some groups have tried to identify eukaryotic activators with reduced toxicity (Baron *et al*, 1997; Khalil *et al*, 2012). Still, we would recommend avoiding long-term use of common eukaryotic activators (utilizing VP16, VP64, or GAL activator domains, including in dCas9-, TALE-, or zinc finger-based synthetic regulators) until their genetic stability has been carefully tested in long-term evolution experiments. Our experiments could be considered as testing rtTA activator stability in various environments. The experiments revealed the evolutionary instability of rtTA, but also led to the discovery of mutant activators and gene circuit designs with lower activator toxicity. These could become novel parts and designs minimizing activator toxicity when eukaryotic activators are needed, as in memory circuits (Ajo-Franklin *et al*, 2007; Burrill *et al*, 2012).

To conclude, this work highlights the unique ability of synthetic biological constructs to provide improved, quantitative understanding and predictability to fundamental biological processes such as evolution and development. Similar studies will be essential to assess and improve the evolutionary stability of synthetic gene circuits, enabling their industrial and clinical application. Therefore, synthetic biology is about to reverse the information flow toward other fields of biology, the source of original inspiration for parts and concepts for the first synthetic genetic constructs.

# Materials and Methods

### Strains and media

We used the haploid *Saccharomyces cerevisiae* strain YPH500 (α, *ura3-52, lys2-801, ade2-101, trp1Δ63, his3Δ200, leu2Δ1*; Stratagene, La Jolla, CA) with the PF synthetic gene circuit stably integrated into chromosome XV near the *HIS3* locus as described previously (Nevozhay *et al*, 2012). Cultures were grown in synthetic dropout

(SD) medium with 2% weight of sugar (glucose or galactose) and the appropriate supplements (-*his*, -*trp*) to maintain auxotrophic selection (reagents from Sigma, St. Louis, MO).

### Experimental evolution

In preparation for the experiments, the PF ancestor strain was streaked on SD 2% glucose plates. Plates were incubated at 30°C for 2 days. Well-isolated single colonies were picked into 1 ml SD-*his-trp* 2% galactose liquid medium and incubated overnight at 30°C with orbital shaking at 250 rpm and resuspended regularly (every 12 h or every 24 h). Fluorescence and cell density measurements were taken daily or every 12 h. Samples were saved daily and stored in 80% glycerol at −80°C for further studies. Further details are described in the Appendix.

### Fitness landscape mapping and parameter estimation

Ancestral PF cells were prepared as described above. Cultures were then resuspended into the following treatments: zeocin only (0.5, 1.0, 1.5 and 2.0 mg/ml), doxycycline only (0.2, 0.5, 1 and 2 µg/ml), and both doxycycline (0.2, 0.5, 1 and 2 µg/ml) and zeocin (2 mg/ml). Cell density and fluorescence were measured every 6 h over 72 h.

Population and cellular growth rates were estimated using mathematical models described previously (Nevozhay *et al*, 2012) and as described in the Appendix. Briefly, we used fitness functions to model the effects of conditions and gene expression on growth. One depends on zeocin and yEGFP::ZeoR protein concentration: $\gamma_1 = \frac{\chi}{\chi + Z_i(F,Z)}$ where $Z_i$ is inferred from the steady-state solution of a dynamical model:

$$\dot{Z}_i = \phi Z - h_z Z_i - sRZ_i$$
$$\dot{B} = sRZ_i - dB$$

with $Z$, $B$, and $R$ representing external zeocin, and bound and unbound yEGFP::ZeoR protein concentrations ($F = B + R$). The other depends on doxycycline and yEGFP::ZeoR protein concentration, assumed to be equal with rtTA protein concentration: $\gamma_2 = g_0 \frac{\alpha}{\alpha + F \frac{C}{C+\beta}}$ with $C$ representing doxycycline concentration. The total growth rate is then $\gamma = \gamma_1 \gamma_2$.

In each condition, the rate of switching from low to high expression and vice versa (cellular memory) was inferred from experimental dose responses in doxycycline as described previously (Nevozhay *et al*, 2012; Appendix Fig S3E).

Resulting parameter estimates are presented in Appendix Table S1.

### Statistical analysis of gene expression and fitness data

Fluorescence and fitness values were compared using *t*-tests in our study. We used an "independent samples" version of the *t*-test to compare different conditions (for example, D0Z0 and D0Z2). On the other hand, we used a dependent (paired) samples version of the *t*-test to compare different time points within one environmental condition. We applied Bonferroni correction for multiple comparisons whenever applicable. All tests were performed in STATISTICA 9.1 (StatSoft Inc., Tulsa, OK).

### Gene expression and fitness characterization of clonal isolates (phenotyping)

Fitness of clones isolated from evolved populations was estimated using an Infinite M200 Pro plate reader (Tecan) for $OD_{600}$ measurements ($600 \pm 9$ nm, number of reads = 25) of orbitally shaken (280.8 rpm with amplitude 2 mm) 250 µl cultures in 96-well plates at $30 \pm 0.5°C$. Cultures were rediluted into fresh media of identical composition every 12 h. Fluorescence was measured every 24 h by flow cytometry.

### Mathematical and computational models

We have developed two different types of predictive models. The first model was a set of ordinary differential equations (ODEs) with the number of ancestral cells, and mutants lumped into K, T, G categories as variables, assuming constant population size. A detailed description of the model is in the Appendix.

The second model was an evolutionary simulation framework explicitly accounting for each individual mutation over the time course written in Python 3. In the framework, we used a linear system of ordinary differential equations (ODEs) to describe ancestral and mutant cells, with experimentally inferred growth rates ($g_L$, $g_H$) and switching rates ($r$, $f$). The simulation framework includes population growth, zeocin internalization dynamics, entry of mutation types K, T, or G into the population and simulated 12-h resuspensions (Appendix Fig S1B and C). We simulated all of the experimental conditions with appropriate growth and switching parameters. To test the effect of preexisting mutations, we simulated a 24-h period without selection before changing the parameters to those appropriate for each condition. The simulation framework is described in greater detail in the Appendix along with its Python script.

The rates of switching, growth, and zeocin internalization were all determined experimentally prior to the simulations (Appendix Table S1). Thus, the three free parameters in each condition were beneficial mutation rate (µ), and the relative probabilities of a mutation being of type K, or T. These parameters were systematically scanned in both models to determine the robustness of our predictions.

### Mutation time course reconstruction

We reconstructed time courses of mutation frequencies for experimental evolution replicates D2Z0-12 h-1 (Fig 3D), DiZ0-12 h-1 (Fig 3E), D0Z2-12 h-1 (Fig 4B), D2Z2-12 h-1 (Fig 5D), DiZ2-12 h-1 (Fig 5E), D2Z0-24 h-1 (Appendix Fig S4B), D0Z2-24 h-1 (Fig 4B, Appendix Fig S5B), and D2Z2-24 h-1 (Appendix Fig S6B). In each case, we had allele frequencies inferred from either Sanger sequencing alone, whole-genome Illumina sequencing alone, or both. For time points with allele frequency data from both methods, we plotted the mean of frequencies between the methods. If only one sequencing method was applied for a given time point, we used the corresponding inferred allele frequency. The number of sequenced time points varied between 2 and 6 depending on the condition (excluding $t = 0$ h; gray points). Once the ancestral genotype reached 0%, we kept it at 0% (even if in rare cases mutant alleles could not account for 100% of the population afterward).

In many conditions, we observed multiple mutations in the same sample. To obtain information on linkage, we performed Sanger sequencing on clonal isolates, at mutation loci determined by whole-genome Illumina sequencing. When linked mutations were called, we averaged the whole-genome frequency estimates of the two mutants to approximate the linked allele frequency. We then averaged that value with frequency estimates from Sanger sequencing. This method permitted inference of linked-mutant frequencies at time points that used both sequencing methods. For example, Appendix Fig S4B shows just the whole-genome inferred allele frequencies for all whole-genome-sequenced time points with no linkage data (thus erroneously indicating lack of linkage for all detected alleles). To illustrate the likely course of allele dynamics at times between the measured points, we used second-order spline interpolation (gray lines).

**Expanded View**  for this article is available online: http://msb.embopress.org

### Acknowledgements

This research was supported by the NIH Director's New Innovator Award Program (1DP2 OD006481-01), by NSF/IOS 1021675 and the Laufer Center for Physical & Quantitative Biology to GB and an Alfred P. Sloan Research Fellowship to AVM. DN acknowledges support from Program # 1326 of the Ministry of Education and Science, Russian Federation, and CG acknowledges support from the Division of Academic Affairs at the MD Anderson Cancer. Sequencing was performed at MD Anderson's DNA Analysis core facility (funded by NCI CA16672). We would like to thank the organizers and participants of the NSF-supported (Grant #1066293) Aspen Center for Physics workshop "Evolutionary Dynamics and Information Hierarchies in Biological Systems" (2012) and the NSF-supported (Grant #PHY11-25915) "Cooperation and the Evolution of Multicellularity" workshop (2013, Kavli Institute for Theoretical Physics) for discussions. MM would like to thank Bill Flynn and Ariella Sasson for assistance with sequencing data analysis. We thank J. Xavier, M. Rosner, D. Charlebois, M. Szenk, T. Székely, S. Levy and J. J. Collins for comments and discussions. We also thank two anonymous reviewers for their highly insightful and constructive comments.

### Author contributions

CG, JCJR, MM, RMA, DN, AVM, and GB designed research; CG and DN performed experiments; CG, JCJR, MM, RMA, DN, and GB analyzed the data; CG, JCJR, MM, RMA, DN, AVM, and GB developed computational models; and CG, JCJR, MM, RMA, DN, AVM, and GB wrote the paper.

### Conflict of interest

The authors declare that they have no conflict of interest.

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
