## [Review Process File · Molecular Systems Biology]

Stress-response balance drives the evolution of a network module and its host genome

Caleb González, J. Christian J. Ray, Michael Manhart, Mr. Rhys M. Adams, Dmitry Nevozhay, Alexandre V. Morozov and Gábor Balázsi

Corresponding author: Gábor Balázsi, Stony Brook University

Review timeline:	Submission date:	21 March 2015
	Editorial Decision:	20 April 2015
	Revision received:	15 July 2015
	Editorial Decision:	20 July 2015
	Revision received:	25 July 2015
	Accepted:	04 August 2015

Transaction Report:

Editor: Maria Polychronidou

1st Editorial Decision

20 April 2015

Thank you again for submitting your work to Molecular Systems Biology. We have now heard back from the two referees who agreed to evaluate your manuscript. As you will see from the reports below, the referees acknowledge that the presented findings seem potentially interesting. However, they raise a series of concerns, which should be carefully addressed in a revision of the manuscript.

The referees' recommendations are clear and therefore there is no need to repeat the points listed below. We would like however to raise your attention to some of the more fundamental issues raised:

- Reviewer #2 refers to the need to perform additional analyses to better support the main findings and provides constructive suggestions in this regard.
- Reviewer #1 mentions that statistical support should be provided for some of the presented findings.
- Both reviewers point out several cases of figure panels and parts of the text that need to be improved for clarity.

Moreover, as referee #1 suggests, we think that a discussion on the broader relevance of the presented approach and analyses should be included.

On a more editorial level, we would like to mention that while we generally encourage the submission of individual Supplementary Figure/Table files, in

exceptional cases (and depending on the nature of information provided) we allow the use of a single PDF file including a Table of Contents. We think that in this case, it presenting the Supplementary Information in a single PDF seems more appropriate.

If you feel you can satisfactorily deal with these points and those listed by the referees, you may wish to submit a revised version of your manuscript. Please attach a covering letter giving details of the way in which you have handled each of the points raised by the referees. A revised manuscript will be once again subject to review and you probably understand that we can give you no guarantee at this stage that the eventual outcome will be favorable.

Reviewer #1:

Summary

In this study, the authors used a previously developed synthetic gene circuit to address a simple question: How does the evolutionary dynamics of a population depend on the environment? To address this question, they built mathematical models to account for the potential invasion of three types of mutants: knockout (K), tweaking (T), and extra-circuit or genomic (G).

Their circuit consists of a dox-inducible positive feedback control of a gene conferring zeocin resistance. Activation of the circuit is tightly coupled with cellular fitness. In particular, circuit activation comes with a cost due to the toxicity of rtTA, but has benefits when cells are cultured in the presence of zeocin. Despite this complex interplay among the circuit activation, cellular fitness, and the environmental stress level, their modeling was able to predict the outcome of evolutionary dynamics in different environmental conditions. They then validated their model predictions by extensive evolutionary experiments, coupled with sequencing analysis.

General comments

This exciting, significant piece of work addresses an important question in evolutionary biology. The analysis is thorough and comprehensive, with overall convincing conclusions. I would strongly endorse its publication in MSB with some revision. The questions I have are mostly focused on the clarification of results or some conceptual issues.

While it is focused on the analysis of a synthetic gene circuit, the methodology (modeling and experiments) is generally applicable to the analysis of evolutionary dynamics in other contexts, such as evolution of antibiotic-resistant bacteria or drug-resistant cancer cells. At the highest level, one might argue that it's expected that environmental condition would dictate the evolutionary dynamics of a population. Moreover, a particular genotype would fare better in certain

environments than in others. For each case, it is perhaps not difficult to make predictions on the evolutionary dynamics by constructing mathematical models. However, it is non-trivial task to test such model predictions and draw definitive conclusions.

In major part due to the use of a well-defined gene circuit, the authors were able to provide highly specific predictions, based on the measurements of some fundamental parameters (e.g. how the cellular fitness depends on gene expression levels - Fig 1B), and then thoroughly tested the predictions experimentally. It is quite remarkable how well their experimental results match their model predictions.

Related to this point, I'm interested in hearing the authors' further clarifications on two questions:

- 1) In general, to predict the evolutionary dynamics of a circuit (or a network), what are the most essential parameters to measure a priori?
- 2) Given their analysis on the synthetic positive feedback circuit, what are the mechanistic insights that are generally applicable (beyond the specific circuit)?

Addressing these questions might help to better clarify the conceptual significance of the work.

Other specific points

1. The work focuses on examining the impact of the environment on the emergence/dominance of certain variants of a gene circuit. It might be worthwhile to comment on whether results presented here can help guide engineering of more robust gene circuits (i.e., more resistant to invasion of mutants). A major factor contributing to this dynamics is the coupling between the circuit activation state and cellular fitness. To this end, the authors might want to note a couple of relevant studies:

Wu et al 2014, Quorum-sensing crosstalk-driven synthetic circuits: from unimodality to trimodality, *Chem & Biol*.

Tan et al 2009, Emergent bistability by a growth-modulating positive feedback circuit. *Nat Chem Biol*.

2. The paper presented tremendous amount of data and analysis. Presentation of some of these can be improved/clarified. For example, while I got the basic concept of Figure 1B, I thought it was somewhat hard to digest. The 3D-surface plot is not easy to gauge relative levels. For the condition, there seems to be 3 dots - not sure why that's the case. For individual panels, what's the scale of x-axis (gene expression)? I assume it's the log of some values but it should be clearly explained.

3. Still for Figure 1B: for a fixed zeocin, my first response was that the cellular fitness dependence on gene expression should be the same for all different dox levels. But that's not the case and it is somewhat surprising to me. It would be helpful if the authors can clarify the intuition behind the difference between the different cellular fitness landscapes.

4. The authors might consider revising the right hand side of Figure 2A to better illustrate their simulation scheme. Right now, the right hand side is not quite informative.
5. On page 7 bottom, the three probabilities ($P(K)$, $P(G)$, and $P(T)$) add up to 1.0. The authors should clarify that the total fraction of mutants is much less than one (they noted this in the figure legend). Overall, it'll be helpful to provide more details on the simulation algorithm in the main text. For example, they might consider illustrating the algorithm using a diagram, as part of Figure 2A.
6. Figure 2B, C: I'm a bit lost on the rationale of looking at the dependence of different allele frequencies on $P(T)$. The authors stated that "... suggesting a possibility of to estimate $P(T)$ and $P(K)$...". But how? Now that it's presented, it would be helpful to explain the trends more clearly. For instance, in Figure 2C, why does the ancestral half-life increases with increasing $p(T)$? Also, when $P(T)$ is varied, what happens to $P(K)$ and $P(G)$? Do Figure 2B and 2C look the same regardless of how $P(K)$ and $P(G)$ are changed, as long as $P(K)+P(G) = 1-P(T)$? Also for the corresponding text on page 8, the authors may be referring to Figs 2B,C, instead of Figs 2B,D.
7. The in-text description of some figures did not appear to match the figures. In particular:
- Fig 4A- Text claims the fluorescence expression increased noticeably; however, the difference between the control and the +antibiotic is not huge (particularly for the top panel). Could you run some statistics to confirm the significance of the difference?
 - Fig 5A-Text claims that the average fluorescence decreased while fitness increased. Just looking at the graph, the decrease in fluorescence does not look significant. Also, the fitness trend would appear to be better described as "starting out high, decreasing temporarily, and then returning to its original level".
 - Fig 5B- The figure doesn't appear to support the text. "...fitness decreased...afterwards fluorescence decreased slowly..." From the figure, fitness briefly decreases- the current phrasing makes it sound like it is decreasing over the entire window of observation.
- For all these, the authors might consider using linear scale for the fluorescence, which may better reflect the trends they were claiming.
8. Page 11 bottom: the authors claimed that "... This effect is PF-dependent, since cells lacking the gene circuit do not survive in D0Z2". The statement might be true but the data are insufficient to support it. The data only shows that cells lacking ZeoR (PX cells) didn't survive in zeocin - in fact, these cells still carry the PF module according to Figure E4C. A more likely interpretation is that the cells should start with some basal level of zeocin resistance, whether or not this is regulated by a PF.

Reviewer #2:

A. Summary of the paper

Gonzalez et al. aimed to quantitatively address the question: "How can a closed genetic circuit that doesn't interact with any genes that are outside the circuit, and which has an intrinsic trade-off in fitness (cost of making a resistance protein vs. protection conferred by the protein), evolve under a selection pressure (presence of antibiotics: doxycycline & Zeocin)?" To address this question, the authors used the budding yeast and a simple yet interesting genetic circuit composed of exogenous genes. Their circuit consisted of two genes. One encoded "rtTA", a transcription factor that binds to TET promoters with doxycycline's help. When rtTA binds to a TET promoter, it increases the expression level of the gene controlled by that TET promoter. Another gene encoded the protein "ZeoR" fused to GFP, which protects the yeast from the antibiotic "Zeocin". TET promoters controlled both the rtTA and ZeoR::GFP expressions. Because rtTA bound to doxycycline is harmful to the cell while the simultaneously produced ZeoR protects the cell from Zeocin, this circuit poses an unavoidable trade-off in fitness. The authors tested how this trade-off affected the evolution of the circuit when the cells were subjected to different combinations of doxycycline and Zeocin. To test this, the authors performed serial dilutions of the yeasts containing the synthetic circuit over many days (~20 days). They sequenced whole populations and measured expression level of ZeoR::GFP in some of the intervening days. The authors found that mutations within and outside the circuit occurred at different frequencies as a function of the type of fitness landscape imposed by the combinations of the two antibiotics. These mutations allowed the population to fully recover within the 20 days. The authors present a mathematical model in the extended view to complement their experiments.

B. Overall evaluation - Accept with a major revision

I think the most interesting and important aspect of the manuscript, and one that should be highlighted more thoroughly, is the evolution of a genetic circuit that imposes a trade-off in the cell's fitness (as seen in Fig. 1B: DiZ2 and D2Z2). But this aspect is obscured by a number of other messages that the authors are trying to simultaneously tell us, which are not very convincing and at times incorrect. There are also several logical gaps in the authors' mathematical model and experiments. By correcting these and by elaborating more on the main aspect mentioned above through a significant revision of their presentation (e.g., many plots are confusing and not explained) and some (but not necessarily all) of the experiments and models that I ask for in the sections below, I think the revised manuscript could be published in *Molecular Systems Biology*.

The most interesting aspect of the manuscript is that the authors show that the yeast evolves to the peak of a fitness landscape by using the extra degree of freedom, namely mutations outside the genetic circuit that are not subject to the trade-off, along with mutations within the circuit (e.g. rtTA) that are subject to the trade-off. What is interesting here is that although mutations outside the synthetic circuit has the benefit of potentially increasing the cell's fitness, it's a more indirect way of dealing with the stress (i.e., presence of the antibiotic) than tuning the TET promoter within the circuit to change the expression level of the antibiotic resistance gene. There are no guarantees that mutations outside the circuit would help the cells resist the antibiotic. Moreover, one would intuitively think that it will

take a "longer time" to find those mutations by randomly mutating sequences outside the circuit. On the other hand, mutations within the circuit would increase the amount of harmful "baggage" for the cell (i.e., the rtTA expression level) despite being the most direct way of resisting the antibiotic. Thus there is not just a trade-off within the circuit, there's also a trade off in having beneficial mutations in the genome outside the circuit and having them within the circuit. This type of "multi-layered" trade-off occurs frequently, not just in synthetic circuits in yeast, but in many natural and more complex genetic circuits in almost every organism. The authors' clever design of their synthetic circuit enabled them to precisely measure and pinpoint to where the different types of trade-offs are. That is the benefit of the synthetic circuit (not that it's a "newcomer" circuit, more on this in Section C). That makes the authors' work a very fascinating systems-level case study of how a circuit evolves by balancing different trade-offs. Therefore I think this manuscript, after a major revision, should be published in *Molecular Systems Biology*.

C. Major comments

C1. The authors emphasize the "bet-hedging" aspect of their circuit (i.e. the bimodal expression of ZeoR) throughout the manuscript. But the term "bet-hedging" is truly meaningful if the environment is fluctuating (e.g., antibiotic is present, then removed, then present again). This way, the population with a bimodal gene expression is ready to deal with both environments at the same time. But in all the experiments, the authors don't fluctuate the environments in this way (which can be done, for example, by diluting the cells into media without Zeocin for one day, and then diluting the cells into a media with Zeocin the next day, and so on). All their experiments deal with fixed environments (e.g., antibiotic is always present or always absent). So the importance of the bimodal expression of ZeoR is unclear. DiZ0 and DiZ2 are the only tested conditions in which there is a bimodal expression of ZeoR (Fig. 1B). But in all three replicates of DiZ0, there is virtually no difference in the fitness of the control (D0Z0, crosses in Fig. 3B) and those under "stress" (DiZ0, blue squares in Fig. 3B). So bimodality seems to have no role in DiZ0 (in fact, DiZ0 is basically like the control). In DiZ2, the fitness between the control and the stressed populations are more different. Thus DiZ2 the only tested situation in which the bimodality might matter. But even in DiZ2 the difference in the growth rates of DiZ2 cells and of the control (D0Z0) (Fig. 5B) is still quite smaller than the difference between the fitness of D2Z2 cells (with unimodal expression level of ZeoR) and that of the control cells. In fact, the lowest growth rate reached in DiZ2 (lowest value of the green squares in Fig. 5B; around Day 5 in all replicates) and the "recovered" growth rate (highest values of the green squares in Fig. 5B) are virtually identical. So it's difficult to see how bimodality affects the evolution in any of the conditions tested. One thing the authors can do is increase the doxycycline just above the level used for DiZx. That will change the fraction of ON VS. OFF cells in a population. Then, by changing this fraction, the authors can show what role bimodal gene expression has on the evolution of the circuit. If it doesn't play a role, then the authors should note it and not emphasize it in the manuscript because doing so detracts from the main message.

C2. The mathematical model doesn't seem to have the predictive power that the authors claim. I think this is the weakest part of the manuscript. My main problem with the model is that the model's predictions are too broad to be falsifiable by

experiments. For example, the model doesn't predict the exact genes that would mutate (this is admittedly unrealistic). It just says that there will be mutations in the circuit or not. The model (equations on Pg 11 of extended view) consists of many equations with many fit parameters (Table E1). It's difficult to see how these lead to a precise prediction rather than finding the parameter values that still yields what one wants to see in the end (results of the experiment). Moreover, none of the outcomes in the evolution experiments were really counter-intuitive to warrant a mathematical model (so the model seems unnecessary and an overkill for the experiments). Perhaps this is my misunderstanding. But in that case, the authors should at least outline their model in the main text in a more substantial (but still intuitively understandable) manner. Right now, it's all in the expanded view section. The description of the model in the main text doesn't have sufficient details to understand Fig. 2 (model-produced results).

C3. What I found most surprising and interesting was that the evolution occurred fairly quickly. In all cultures, evolved strains had a measurable effect on the growth rate of the entire population within 2-3 days after the selection pressure (doxycycline or Zeocin) was applied (Figs. 3 & 4). Is it possible that the beneficial mutants existed as very few cells at $t=0$? Such a small number of mutants that already exist in the starting culture would escape detection. It's not experimentally possible to check that these mutants existed if they were there as a very tiny fraction of the population. But one can construct a mathematical model to see what effect such a pre-existing small population of mutants would have (e.g., you can calculate typical time you need to wait for a mutation to occur, and compare that with the time you'd expect to wait for the same type of mutations to come up). That's where a model might be beneficial. If this scenario isn't true, then it would be helpful to a reader if the authors can provide some insights into why there is a high enough fraction of evolved strains in the population to increase the overall fitness of the population in these early time points.

C4. One possible outcome that hasn't been observed is $rtTA$ mutating into tTA . tTA is the reverse of $rtTA$ in that it activates a gene driven by TET02 promoter in the absence of doxycycline and represses the gene when doxycycline is present. These mutations are known in the literature. This can clearly rescue the cells' fitness in DiZ0 and D2Z0. The fact that this wasn't observed is puzzling. Some insight into why this wasn't observed would be helpful.

C5. The authors only plated and picked colonies of evolved strains from the 21st day (last day) of growth in each experiment (Figure 6). It's unclear that these strains actually represent the genotypes and phenotypes that exist in the populations during their most interesting recovery time (days 5-10 in all their evolution experiments). To get a sense for how many different strains and in what proportions the different strains are present within a single culture, the authors should consider plating the evolving cultures during the intense recovery phase (one of the days within Day 5 to Day 10) in which the growth rate really starts to go back up from the local minimum.

C6. Related to above, it's not clear to me that the "allele frequencies" obtained from sequencing the entire liquid culture (without isolating single colonies) at different days tells you about the fraction of different mutants present in a population. For

example, in Fig. 3C & 3D, is the allele frequency telling us the fraction of cells within a population that has that allele? Does this method distinguish a scenario in which two mutations are present in the same cell from a scenario in which the two mutations are in two different cells? By plating liquid culture at different days (particularly at one of the days of sharp recovery: e.g., Day 5 & 6 in Replicate #2 in Fig. 3A), then picking individual colonies, sequencing them and characterizing their phenotypes, one could answer these types of questions. If I'm mistaken in my logic here, then the authors should more clearly explain what the allele frequency is telling us in terms of number of cell types that are present in one liquid culture over time.

C7. Figure 3: It's surprising that no mutations outside rtTA were observed even after 19 days. Were there not even neutral mutations outside the circuit? How is the rate of neutral mutation inferred from Fig 3 (apparently very rare over 19 days) compatible with the seemingly more frequent rate of mutation in Fig. 4 & 5? This should be clearly explained in the text and the model.

C8. Comparing results from experiments in which dilution was done every 12-hours (Fig. 3-5) with those in which dilution was done every 24-hours (Extended view Figs. E3-E5), the mutations are quite different. For example, Fig. E3-B and Fig. 3D test the same D2Z0, but the mutations do not overlap in the two cases. Moreover, the population's growth rate seems to recover more quickly in the 24-hr resuspensions (in extended view) than in the 12-hr resuspensions (main figures). But the authors mention that they are basically the same. Please state more clearly how the dilution rate affects the evolution experiments.

C9. Related to point C3, the authors can stress more what's conceptually so important about the interplay between mutations that occur in their circuit and those occurring outside of the circuit. Can this be related to the time one would need to wait for a cell to find a beneficial mutation outside of the circuit? That is, what is the expected number of beneficial hits per unit time within the circuit and outside the circuit? Perhaps addressing this point gets to the heart of the trade-off between mutations in and outside the circuit. If the authors can provide a conceptual insight on this point through their experiments or a model, it would significantly improve this manuscript.

C10. One of messages that the authors stress is that they're studying how a "newcomer" network evolves. As they mention, it's true that a de novo network evolves quickly through horizontal gene transfer or recombination. But what is so special about being a newcomer isn't clear. Instead, they are really looking at a close network's evolution through mutations in itself and outside. There is no dotted line that says the circuit is new, and the cell certainly can't tell what is a "newcomer" circuit and what isn't. The only distinction that makes sense is whether you have a close network or not. So this focus on a "newcomer" network throughout the manuscript isn't clear to me.

C11. The current presentation (both text and figures) detracts from what I think is a thoughtful and interesting study. The presentation would benefit from a substantial revision. Many of the plots in the main figures are not self-explanatory and not sufficiently explained in the figure captions. For example, it took me a long time

just to figure out what the different colors or the numbers were in Fig. 3D and how those smooth lines were drawn there despite measuring only 3 time points (these are not explained in the captions, in the main text, or in the figure). The main text consists of many long sentences that contain unnecessary, and sometimes incorrectly applied technical jargon. Shortening these sentences, and using just simple basic words will clarify the authors' main message.

D. Minor comments

D1 - Many sentences were often too long and contained too many jargon. These made it difficult to understand the story at times. Often, no jargon would have been necessary to explain the experimental results.

D2 - Figure 1B: Uses both "Cellular fitness" and "Population fitness". Cellular fitness is computed from the mathematical model. It's used to compute the population fitness. But the population fitness landscape in Fig. 1B should be measured, not calculated.

D3 - Figure 2A: Cartoon of dividing cells is confusing. Also, which category do mutations in ZeoR ORF fit in? K, T, and G seem only applicable to the rtTA or the TET promoters.

D4 - Figure 3A & 3B: The heat shades for the yEGFP measured in a flow cytometer. On some of the days (e.g., D2Z0 days 5-10 in replicate #1 in Fig. 3A), ON and OFF cells coexist in the same liquid culture. Is this because there are cells with bimodal expression (bistable) yEGFP or is it because there is a mixture of uni-modal ON and uni-modal OFF cells in the same liquid culture?

D5 - Figure 3D: Measurements for only 3 of the days (Day 0, 9, 19) but smooth lines joining these points. Not straight lines but lines with curvature. How did you get these lines? From a model or a spline-fitting? Moreover, the colors are not explained in the caption or in the figure.

D6 - Fig. 4B & 5D: Same comment as above.

D7 - Fig. 6 is very confusing. It's unclear what the main message is here.

D8 - Extended view - Fig.E3-B doesn't look qualitatively the same as Fig. 3D, even though they are for the same environmental condition: D2Z0.

Responses to the Reviewers' comments: MSB-15-6185

Reviewer #1:

This exciting, significant piece of work addresses an important question in evolutionary biology. The analysis is thorough and comprehensive, with overall convincing conclusions. I would strongly endorse its publication in MSB with some revision. The questions I have are mostly focused on the clarification of results or some conceptual issues.

Response: We are excited to see that our manuscript made an overall positive impression. We thank the Reviewer for his/her enthusiastic and supportive words, as well as for the insightful comments on our work.

While it is focused on the analysis of a synthetic gene circuit, the methodology (modeling and experiments) is generally applicable to the analysis of evolutionary dynamics in other contexts, such as evolution of antibiotic-resistant bacteria or drug-resistant cancer cells. At the highest level, one might argue that it's expected that environmental condition would dictate the evolutionary dynamics of a population. Moreover, a particular genotype would fare better in certain environments than in others. For each case, it is perhaps not difficult to make predictions on the evolutionary dynamics by constructing mathematical models. However, it is non-trivial task to test such model predictions and draw definitive conclusions.

In major part due to the use of a well-defined gene circuit, the authors were able to provide highly specific predictions, based on the measurements of some fundamental parameters (e.g. how the cellular fitness depends on gene expression levels - Fig 1B), and then thoroughly tested the predictions experimentally. It is quite remarkable how well their experimental results match their model predictions.

Related to this point, I'm interested in hearing the authors' further clarifications on two questions:

1) In general, to predict the evolutionary dynamics of a circuit (or a network), what are the most essential parameters to measure a priori?

Response: Before measuring any parameters, we believe that the following conditions must be satisfied to make short-term evolutionary dynamics of a network (module) predictable. (i) The network should operate relatively independently of the host genome in each environmental condition where its evolution is studied. (ii) Mutations altering the network's dynamics should affect the fitness of the host organism much more than any other mutations. (iii) The network should be sufficiently small and its components should be sufficiently well-characterized to allow the development of quantitative models of network dynamics and its effects on cellular as well as on population fitness. These requirements imply a need to quantitatively connect cellular fitness to network dynamics. To

fulfill condition (iii), typical biochemical parameters, such as gene regulatory functions, degradation, association and dissociation rates are necessary to unravel network dynamics. In addition, the dependence of the cell division rate (cellular fitness) on gene network state must be known. In our case, we characterized the dependence of cellular fitness on regulator and effector protein concentrations. (These should correlate strongly in the PF system because both genes are expressed from the same promoter). More generally, for gene circuits with fast dynamics, such as rapid oscillators, one may need to determine how cell division rate depends on the history of gene expression in single cells. Creating these connections may be non-trivial, especially when cellular growth rates affect protein dilution rates and thereby network dynamics, as described in Tan et al., Nat. Chem. Biol. 5(11):842-8 (2009). Once acquired, such *a priori* knowledge should allow calculating how mutations (corresponding to parameter changes in network models) alter network dynamics, and thereby cellular and population fitness. Consequently, knowing the fitness corresponding to different mutation types allows predicting evolutionary dynamics for various mutation rates. We note that it is sometimes possible and useful to develop coarse-grained models, linking phenotypic switching and growth rates for a few cellular states to specific network components as well as cellular and population fitness.

2) Given their analysis on the synthetic positive feedback circuit, what are the mechanistic insights that are generally applicable (beyond the specific circuit)?

Response: We obtained three main mechanistic insights.

i) The environment dictates which types of mutations are observed in evolution, as long as the mutations are available. Mutations most capable of improving fitness in a given environment will be predominantly observed, even if they are not the most common type to arise.

ii) Drug resistance incurs a cost, even if it evolves *de novo*. For example, in D022 gene expression had to increase while the PF circuit was inactive in the absence of inducer, and beneficial mutations were not readily available inside the gene circuit. Nevertheless, extra-circuit mutations occurred that improved gene expression within the gene circuit. Overall, these mutations elevated yEGFP::zeoR gene expression, causing antibiotic resistance, which was beneficial in the presence of the drug, but – interestingly - costly without the drug. This is an interesting novel tradeoff that we discovered from the evolution experiments.

iii) Many mutations are available that destroy protein (here, rtTA) function. On the other hand, many fewer mutations are available that can fine-tune protein function. Once we select for weaker (but not completely absent) rtTA function, one of the few such mutations will show up. This is probably why we observed identical point mutations at the 225th basepair of rtTA repeatedly in D222. This is consistent with other experiments on microbial evolution under strong selection, such as bacteria under antibiotic stress where the same mutations appeared repeatedly. The paper by Toprak et al. (Nat Genet 44: 101-105, 2012) on antibiotic resistance is striking in this regard. It would be interesting to understand the mechanisms underlying the recurrence of these adaptive changes.

Other specific points

1. The work focuses on examining the impact of the environment on the emergence/dominance of certain variants of a gene circuit. It might be worthwhile to comment on whether results presented here can help guide engineering of more robust gene circuits (i.e., more resistant to invasion of mutants). A major factor contributing to this dynamics is the coupling between the circuit activation state and cellular fitness. To this end, the authors might want to note a couple of relevant studies:

Wu et al 2014, Quorum-sensing crosstalk-driven synthetic circuits: from unimodality to trimodality, *Chem & Biol*. Tan et al 2009, Emergent bistability by a growth-modulating positive feedback circuit. *Nat Chem Biol*.

Response: We thank the Reviewer for these insightful comments. The evolutionary stability of synthetic gene circuits has been examined by the Herbert Sauro group in *Escherichia coli*, see Sleight et al. *Journal of Biological Engineering*, 4:12, 2010, which we are citing. Their main conclusion was that using genetic parts (terminators, promoters) with similar nucleotide sequences in gene circuits decreases their evolutionary stability. As far as we know, this question has not been addressed in eukaryotes. Our work addresses this question in eukaryotes and generates two insights for building evolutionarily robust gene circuits.

The first insight is that eukaryotic activators are not ideal gene circuit components if gene circuit stability is a concern. There is evidence that eukaryotic activators are generally toxic (see Baron U, Gossen M, Bujard H, *Nucleic Acids Res* 25: 2723–2729, 1997), which seems to be true for some prokaryotic components as well, see Tan et al., *Nat. Chem. Biol.* 5(11):842-8 (2009). To address this problem, some groups have tried to identify eukaryotic activators with reduced toxicity (see Baron U, Gossen M, Bujard H, *Nucleic Acids Res.* 25: 2723–2729, 1997 and Khalil AS et al., *Cell* 150(3): 647–658, 2012). Still, our data suggest avoiding the use of common eukaryotic activators (utilizing VP16, VP64, or GAL activator domains, including in dCas9-, TALE- or zinc finger-based synthetic regulators) in long-term applications until their evolutionary stability is carefully tested in evolution experiments. Our evolution experiments could be considered as testing rtTA activator stability in various environments. The experiments revealed the evolutionary instability of rtTA, but also led to the discovery of mutant activators and gene circuit designs with lower activator toxicity. Besides activators, silencing domain-based (e.g., KRAB) repressors may also affect fitness, reducing long-term stability. Therefore, instead of proteins with silencing and activator domains, repressors that sterically occlude polymerase access may be a better choice. In general, a cascade of two repressors can act as an activator. Another alternative would be to use evolved genes or gene circuit designs with lower toxicity instead of the common eukaryotic activators when activators are needed, as in memory circuits, see Burrill et al., *Genes Dev.* 26(13):1486-97, 2012 and Ajo-Franklin et al., *Genes Dev.* 21(18):2271-6 (2007).

The second insight is that eukaryotic gene circuit robustness depends not only on the circuit design, but also on the environment. An appropriate change in the environment (such as counter-selection by

antibiotic, for example in D2Z2 and DiZ2) will prolong gene circuit half-life by effectively suppressing the expansion of cells with mutated circuits in the population.

We thank the Reviewer for recommending the two highly relevant papers. We have cited them in the revised manuscript.

2. The paper presented tremendous amount of data and analysis. Presentation of some of these can be improved/clarified. For example, while I got the basic concept of Figure 1B, I thought it was somewhat hard to digest. The 3D-surface plot is not easy to gauge relative levels. For the condition, there seems to be 3 dots - not sure why that's the case. For individual panels, what's the scale of x-axis (gene expression)? I assume it's the log of some values but it should be clearly explained.

Response: We agree and thank the Reviewer for pointing this out. We have plotted the 3D surface such that the relative levels are easier to see. The three dots corresponded to triplicate experimental fitness measurements in each of the six conditions. They were moved from the main text to the Expanded View for clarity. We have clarified in the figure legend that the numbers correspond to $\log_{10}(\text{fluorescence})$ in arbitrary units as in all subsequent plots.

3. Still for Figure 1B: for a fixed zeocin, my first response was that the cellular fitness dependence on gene expression should be the same for all different dox levels. But that's not the case and it is somewhat surprising to me. It would be helpful if the authors can clarify the intuition behind the difference between the different cellular fitness landscapes.

Response: Cellular fitness results from two contributions: Zeocin effect and Doxycycline effect. We regret being unclear previously about the Doxycycline effect when Zeocin is present. Now we explain these fitness contributions in detail in the Expanded View. Briefly, cellular fitness (with or without Zeocin) depends not only on rtTA expression, but also explicitly on the amount of Doxycycline present. This is because the Doxycycline-bound form of rtTA (that can also bind DNA) is more toxic than the free (unbound) form of rtTA. We think that target gene expression saturates (reaches its maximum) while a substantial fraction of intracellular rtTA may still be unbound by doxycycline. Therefore, increasing doxycycline concentration further can cause more toxicity, and more stability of the high expression state without increasing gene expression (fluorescence). Thus, cells with the same total rtTA expression will grow slower and remain in the high expression state longer in high Doxycycline concentration, presumably because more of their rtTA will spend time bound to DNA. This is true whether Zeocin is present or not. More details on this can be found in our earlier work characterizing the PF gene circuit [Nevozhay D, Adams RM, et al. Mapping the environmental fitness landscape of a synthetic gene circuit. PLoS Comput Biol. 8(4):e1002480, 2012]. There we examined quantitatively the dependence of cellular fitness on gene expression, Doxycycline and Zeocin.

4. The authors might consider revising the right hand side of Figure 2A to better illustrate their simulation scheme. Right now, the right hand side is not quite informative.

Response: We appreciate this suggestion. We have improved the right-hand side of Figure 2A as suggested, and we hope that it is now more informative.

5. On page 7 bottom, the three probabilities (P(K), P(G), and P(T)) add up to 1.0. The authors should clarify that the total fraction of mutants is much less than one (they noted this in the figure legend). Overall, it'll be helpful to provide more details on the simulation algorithm in the main text. For example, they might consider illustrating the algorithm using a diagram, as part of Figure 2A.

Response: We have improved the description of the simulation framework in the main text, and clarified the point about the total fraction of mutants in the main text as well. The new version of Fig. 2A is hopefully more illustrative of the algorithm. We have also created a diagram that can be found in the Expanded View, Fig. E1C.

6. Figure 2B, C: I'm a bit lost on the rationale of looking at the dependence of different allele frequencies on P(T). The authors stated that "... suggesting a possibility of to estimate P(T) and P(K)...". But how? Now that it's presented, it would be helpful to explain the trends more clearly. For instance, in Figure 2C, why does the ancestral half-life increase with increasing p(T)? Also, when P(T) is varied, what happens to P(K) and P(G)? Do Figure 2B and 2C look the same regardless of how P(K) and P(G) are changed, as long as $P(K)+P(G) = 1-P(T)$? Also for the corresponding text on page 8, the authors may be referring to Figs 2B,C, instead of Figs 2B,D.

Response: We appreciate the point that the description of this result was confusing in the main text. We scanned the parameter space for all 3 free parameters: μ , P(T) and P(G). This resulted in a large amount of data. For the sake of clarity, we show only the effects of changing a single parameter, P(T), in main text Figure 2, while most effects of the parameter changes are shown in the Expanded View. Therefore, in Figure 2, panels B and C we fixed the values of $\mu_{-Z} = 10^{-6.2}$, $\mu_{+Z} = 10^{-5.4}$, and $P(G) = 0.75$, and then scanned only P(T), implying that $P(K) = 1 - P(G) - P(T) = 0.25 - P(T)$. The effects of varying P(G) and μ are separately shown in the Expanded View (Figs. S2, S3).

The ancestral half-life in condition D2Z0 indeed increases with P(T) in Fig. 2C. On the other hand, it decreases with P(T) in conditions D2Z2 and DiZ2. To understand these effects, it is important to note that in Fig. 2 we fix P(G) and μ . Thus, increasing P(T) will be equivalent to decreasing $P(K) = 0.25 - P(T)$ as explained above. Or, the higher P(T), the lower P(K). The different dependencies of the ancestral half-life on P(T) will result from K mutations being more beneficial than T mutations in D2Z0. This is because K mutations completely eliminate rtTA toxicity while T mutations may only weaken rtTA toxicity. Consequently, K mutations will establish and spread faster than T mutations in D2Z0. Therefore, lower availability of K mutations as we increase P(T) causes increased ancestral half-life in D2Z0. On the contrary, in conditions DiZ2 and D2Z2, T mutations are beneficial while K mutations are deleterious. This is because K mutations completely eliminate drug resistance while T mutations do not. Consequently, the higher availability of T mutations as we increase P(T) causes shorter ancestral PF half-life in DiZ2 and D2Z2.

7. The in-text description of some figures did not appear to match the figures. In particular:

-Fig 4A- Text claims the fluorescence expression increased noticeably; however, the difference between the control and the +antibiotic is not huge (particularly for the top panel). Could you run some statistics to confirm the significance of the difference?

Response: Thank you for pointing out that some gene expression changes were not visually obvious. First of all, we would like to mention that we used logarithmic binning/scale for fluorescence values in all relevant figures (specifically, 3A, 3B, 4A, 5A, 5B). Thus, though it might look small, even a slight shift on the plot can translate into a large numerical difference. However, we agree that this might not be immediately apparent from the plot, therefore we performed statistical analysis, comparing triplicate gene expression averages and fitness measurements (D0Z0 versus D0Z2), to quantitatively demonstrate gene expression shifts. The analysis (see the Results section on D0Z2 and the legend of Fig. 4 as well as the *StatisticalAnalysis* file) showed statistically significant difference between fluorescent values of D0Z2 versus D0Z0 in virtually all time points, which was still true even after correction of multiple comparisons (particularly towards the end of the experiment). We also performed a similar analysis for growth rates and found a statistically significant difference between D0Z0 and D0Z2 at early time points, indicating the profound toxic effect of Zeocin in D0Z2. As expected, the difference disappeared after the first week, confirming that drug-resistant mutants overtook the population, restoring the growth rate to its ancestral level.

-Fig 5A-Text claims that the average fluorescence decreased while fitness increased. Just looking at the graph, the decrease in fluorescence does not look significant. Also, the fitness trend would appear to be better described as "starting out high, decreasing temporarily, and then returning to its original level".

Response: The response to the previous comment applies here too. The overall decrease in fluorescence values in D2Z2 condition over time is not very large, but noticeable and consistent across all replicates. To test the significance of these changes, we performed statistical analysis, comparing triplicate gene expression averages and fitness measurements between Day 4 and Day 21 of the D2Z2 time course. We used Day 4 instead of Day 1 to represent the beginning of the experiment because cells took ~3 days to establish their rtTA and yEGFP::zeoR expression. Likewise, there was a similar lag before the onset of Zeocin and rtTA expression toxicity. We obtained a statistically significant difference in both fluorescence values and growth rates between Day 4 and Day 21 of the D2Z2 time course. These results are described in the Results section on D2Z2 and the legend of Fig. 5.

-Fig 5B- The figure doesn't appear to support the text. "...fitness decreased...afterwards fluorescence decreased slowly..." From the figure, fitness briefly decreases- the current phrasing makes it sound like it is decreasing over the entire window of observation.

Response: We agree with the Reviewer. Indeed, the initial text in the manuscript was unclear. Once again, we performed statistical analysis, comparing triplicate fitness measurements along the DiZ2 time course. There was no statistically significant difference between the fitness of DiZ2 populations at Day 4 and Day 21 (Day 4 was chosen for the same reasons as above). However, we also compared fitness values between DiZ2 and D0Z0 (control) and found statistically significant difference between them in several the time points, which was still true even after correction for multiple comparisons (particularly in the beginning and towards the end of the experiment).

For all these, the authors might consider using linear scale for the fluorescence, which may better reflect the trends they were claiming.

Response: We tried using a linear scale of fluorescence previously. The problem is that it compresses the low expression peak such that it looks much tighter and taller than the high expression peak, making the latter difficult to see. Moreover, small gene expression shifts (such as in D0Z2) are really difficult to see with the linear bins. For these reasons, we kept the log-binning for fluorescence plots.

8. Page 11 bottom: the authors claimed that "... This effect is PF-dependent, since cells lacking the gene circuit do not survive in D0Z2". The statement might be true but the data are insufficient to support it. The data only shows that cells lacking ZeoR (PX cells) didn't survive in zeocin - in fact, these cells still carry the PF module according to Figure E4C. A more likely interpretation is that the cells should start with some basal level of zeocin resistance, whether or not this is regulated by a PF.

Response: We agree. We have changed the text accordingly.

Reviewer #2:

A. Summary of the paper

Gonzalez et al. aimed to quantitatively address the question: "How can a closed genetic circuit that doesn't interact with any genes that are outside the circuit, and which has an intrinsic trade-off in fitness (cost of making a resistance protein vs. protection conferred by the protein), evolve under a selection pressure (presence of antibiotics: doxycycline & Zeocin)?" To address this question, the authors used the budding yeast and a simple yet interesting genetic circuit composed of exogenous genes. Their circuit consisted of two genes. One encoded "rtTA", a transcription factor that binds to TET promoters with doxycycline's help. When rtTA binds to a TET promoter, it increases the expression level of the gene controlled by that TET promoter. Another gene encoded the protein "ZeoR" fused to GFP, which protects the yeast from the antibiotic "Zeocin". TET promoters controlled both the rtTA and ZeoR::GFP expressions. Because rtTA bound to doxycycline is harmful to the cell while the simultaneously produced ZeoR protects the cell from Zeocin, this circuit poses an unavoidable trade-off in fitness. The authors tested how this trade-off affected the evolution of the circuit when the cells

were subjected to different combinations of doxycycline and Zeocin. To test this, the authors performed serial dilutions of the yeasts containing the synthetic circuit over many days (~20 days). They sequenced whole populations and measured expression level of ZeoR::GFP in some of the intervening days. The authors found that mutations within and outside the circuit occurred at different frequencies as a function of the type of fitness landscape imposed by the combinations of the two antibiotics. These mutations allowed the population to fully recover within the 20 days. The authors present a mathematical model in the extended view to complement their experiments.

B. Overall evaluation - Accept with a major revision

I think the most interesting and important aspect of the manuscript, and one that should be highlighted more thoroughly, is the evolution of a genetic circuit that imposes a trade-off in the cell's fitness (as seen in Fig. 1B: DiZ2 and D2Z2). But this aspect is obscured by a number of other messages that the authors are trying to simultaneously tell us, which are not very convincing and at times incorrect. There are also several logical gaps in the authors' mathematical model and experiments. By correcting these and by elaborating more on the main aspect mentioned above through a significant revision of their presentation (e.g., many plots are confusing and not explained) and some (but not necessarily all) of the experiments and models that I ask for in the sections below, I think the revised manuscript could be published in *Molecular Systems Biology*.

The most interesting aspect of the manuscript is that the authors show that the yeast evolves to the peak of a fitness landscape by using the extra degree of freedom, namely mutations outside the genetic circuit that are not subject to the trade-off, along with mutations within the circuit (e.g. rtTA) that are subject to the trade-off. What is interesting here is that although mutations outside the synthetic circuit has the benefit of potentially increasing the cell's fitness, it's a more indirect way of dealing with the stress (i.e., presence of the antibiotic) than tuning the TET promoter within the circuit to change the expression level of the antibiotic resistance gene. There are no guarantees that mutations outside the circuit would help the cells resist the antibiotic. Moreover, one would intuitively think that it will take a "longer time" to find those mutations by randomly mutating sequences outside the circuit. On the other hand, mutations within the circuit would increase the amount of harmful "baggage" for the cell (i.e., the rtTA expression level) despite being the most direct way of resisting the antibiotic. Thus there is not just a trade-off within the circuit, there's also a trade off in having beneficial mutations in the genome outside the circuit and having them within the circuit. This type of "multi-layered" trade-off occurs frequently, not just in synthetic circuits in yeast, but in many natural and more complex genetic circuits in almost every organism. The authors' clever design of their synthetic circuit enabled them to precisely measure and pinpoint to where the different types of trade-offs are. That is the benefit of the synthetic circuit (not that it's a "newcomer" circuit, more on this in Section C). That makes the authors' work a very fascinating systems-level case study of how a circuit evolves by balancing different trade-offs. Therefore I think this manuscript, after a major revision, should be published in *Molecular Systems Biology*.

Response: We would like to thank the Reviewer for the positive and thoughtful comments and for sharing our fascination with the evolution of this synthetic gene circuit. In particular, we appreciate

the suggestion about the multi-layered trade-offs, which is indeed very interesting. Thus, we tried to emphasize it in the manuscript. In addition, we have tried to clarify and correct our presentation as much as possible. We hope that by making these changes the manuscript has improved substantially, and will be suitable to publish.

C. Major comments

C1. The authors emphasize the "bet-hedging" aspect of their circuit (i.e. the bimodal expression of ZeoR) throughout the manuscript. But the term "bet-hedging" is truly meaningful if the environment is fluctuating (e.g., antibiotic is present, then removed, then present again). This way, the population with a bimodal gene expression is ready to deal with both environments at the same time. But in all the experiments, the authors don't fluctuate the environments in this way (which can be done, for example, by diluting the cells into media without Zeocin for one day, and then diluting the cells into a media with Zeocin the next day, and so on). All their experiments deal with fixed environments (e.g., antibiotic is always present or always absent). So the importance of the bimodal expression of ZeoR is unclear. DiZ0 and DiZ2 are the only tested conditions in which there is a bimodal expression of ZeoR (Fig. 1B). But in all three replicates of DiZ0, there is virtually no difference in the fitness of the control (DOZ0, crosses in Fig. 3B) and those under "stress" (DiZ0, blue squares in Fig. 3B). So bimodality seems to have no role in DiZ0 (in fact, DiZ0 is basically like the control). In DiZ2, the fitness between the control and the stressed populations are more different. Thus DiZ2 the only tested situation in which the bimodality might matter. But even in DiZ2 the difference in the growth rates of DiZ2 cells and of the control (DOZ0) (Fig. 5B) is still quite smaller than the difference between the fitness of D2Z2 cells (with uni-modal expression level of ZeoR) and that of the control cells. In fact, the lowest growth rate reached in DiZ2 (lowest value of the green squares in Fig. 5B; around Day 5 in all replicates) and the "recovered" growth rate (highest values of the green squares in Fig. 5B) are virtually identical. So it's difficult to see how bimodality affects the evolution in any of the conditions tested. One thing the authors can do is increase the doxycycline just above the level used for DiZx. That will change the fraction of ON VS. OFF cells in a population. Then, by changing this fraction, the authors can show what role bimodal gene expression has on the evolution of the circuit. If it doesn't play a role, then the authors should note it and not emphasize it in the manuscript because doing so detracts from the main message.

Response: We agree that bet-hedging and bimodality are not essential to most of our observations. We have now downplayed "bet-hedging" in the manuscript (it is only mentioned once in the Introduction, in a context not directly related to the PF gene circuit). On the other hand, bimodality is a byproduct of having an activator with positive autoregulation. In principle, all of our experiments could have been done using a gene circuit where a regulator controls target gene expression without feedback, causing it to increase unimodally while fitness goes down with the inducer concentration. Such a system might have enabled simpler, more straightforward interpretations of the results. However, there were two important reasons for using the PF system. First, we already had this positive-feedback system with known gene expression and fitness characteristics, which seemed highly useful. Second, we think that positive feedback regulation brings an important feature that is beneficial for our work: together with activator toxicity, it creates sufficiently steep fitness landscapes

to produce fast, predictable effects in evolution experiments. Without positive feedback, the overall amount of activator would have been constant, making activator mutations less beneficial. Last, but not least, some interesting evolutionary changes (such as the two peaks of the histogram approaching each other as they move towards the fitness peak) would have been impossible to observe with unimodal expression. Peaks moving closer to each other in the bimodal distribution represent an interesting way of fitness improvement that has not been observed experimentally before as far as we know.

Regarding the fitness changes in the conditions mentioned we have performed statistical analyses (see the *StatisticalAnalysis* file) and as the Reviewer pointed out, there were no statistically significant fitness differences between the DiZ0 or D0Z0 conditions. We have changed the wording in the manuscript to make this clear. However, statistical analysis revealed a significant difference between fitness values of DiZ2 and D0Z0 conditions in some of the time points, which was still true even after correction for multiple comparisons (particularly in the beginning and towards the end of the experiment). We have also reflected this in the manuscript and figure legend. Finally, we think that discussing the distributions' bimodality separately in individual conditions is not really justified. For example, the bimodal histograms in DiZ0 and DiZ2 are not separable. Instead, they relate directly to each other, because Zeocin selects against low-expressing cells and reshapes the histogram to look less symmetric in DiZ2 compared to DiZ0. Likewise, there is a small subpopulation of low-expressing cells in D2Z0, so the distribution is slightly bimodal even in D2Z0. This tiny subpopulation seemingly disappears in D2Z2, but not because cells do not express at low level. Rather, it disappears because low-expressing cells grow slower in Zeocin. Therefore, the distributions in D2Z0 and D2Z2 are closely related and inseparable. Beneficial mutations can arise in these low-expressing cells, especially because their DNA is vulnerable to Zeocin. In unimodal populations, such drastic cell-cell fitness differences would not exist, and evolutionary dynamics may be different.

C2. The mathematical model doesn't seem to have the predictive power that the authors claim. I think this is the weakest part of the manuscript. My main problem with the model is that the model's predictions are too broad to be falsifiable by experiments. For example, the model doesn't predict the exact genes that would mutate (this is admittedly unrealistic). It just says that there will be mutations in the circuit or not. The model (equations on Pg 11 of extended view) consists of many equations with many fit parameters (Table E1). It's difficult to see how these lead to a precise prediction rather than finding the parameter values that still yields what one wants to see in the end (results of the experiment). Moreover, none of the outcomes in the evolution experiments were really counter-intuitive to warrant a mathematical model (so the model seems unnecessary and an overkill for the experiments). Perhaps this is my misunderstanding. But in that case, the authors should at least outline their model in the main text in a more substantial (but still intuitively understandable) manner. Right now, it's all in the expanded view section. The description of the model in the main text doesn't have sufficient details to understand Fig. 2 (model-produced results).

Response: We thank the Reviewer for this useful criticism. We agree that our model description in the main text was insufficient. In the revised version we have tried to improve it, by explaining the

parameter choices, clarifying the results, and explaining better what they mean. For example, the parameters describing gene circuit dynamics and cellular fitness were estimated from 13 experimental gene expression and fitness measurements and are not truly free parameters. This is described now in detail in the Expanded View. There are three free parameters in any given condition: any two of the mutation type probabilities $P(T)$, $P(K)$, and $P(G)$ that must sum up to 1, and the beneficial mutation rate μ . The rest of the parameters are all based on fitness and gene expression measurements taken before the evolution experiments (Fig. 1B), as described in the Expanded View. Using these parameters, we predict not just what types of mutations establish in each condition, but also the number of competing alleles and the speed at which the ancestral genotype disappears from the population. Moreover, seeking agreement between the modeling results and the experimental data, we could roughly estimate the three free parameters. Some mutation types even suggest the genes where they should occur: for example, K or T mutations would be most expected to occur around *rtTA*. To summarize, our computational findings reveal how fast and by what means a genetic regulatory circuit with inherent trade-off and phenotypic switching undergoes the first steps of evolution. We have expanded the Discussion to clarify this point. Further, we have clarified in the main text that the distinction between the G and non-G mutation types is whether they occur in the regulator *rtTA* or not, so that mutations in non-*rtTA* portions of the circuit (i.e. *ZeoR::yEGFP*) belong to G type mutations. Overall, we think that the simulation framework adds to the paper, and we hope its value is easier to appreciate in this revision.

C3. What I found most surprising and interesting was that the evolution occurred fairly quickly. In all cultures, evolved strains had a measurable effect on the growth rate of the entire population within 2-3 days after the selection pressure (doxycycline or Zeocin) was applied (Figs. 3 & 4). Is it possible that the beneficial mutants existed as very few cells at $t=0$? Such a small number of mutants that already exist in the starting culture would escape detection. It's not experimentally possible to check that these mutants existed if they were there as a very tiny fraction of the population. But one can construct a mathematical model to see what effect such a pre-existing small population of mutants would have (e.g., you can calculate typical time you need to wait for a mutation to occur, and compare that with the time you'd expect to wait for the same type of mutations to come up). That's where a model might be beneficial. If this scenario isn't true, then it would be helpful to a reader if the authors can provide some insights into why there is a high enough fraction of evolved strains in the population to increase the overall fitness of the population in these early time points.

Response: We thank the Reviewer for raising this point. In our experiments we grow the cells for a day in *D0Z0* before starting them in the environments where they evolve. So indeed, the chance exists that neutral mutations appear during these 24 hours that will become subsequently beneficial. We have used our computational model to investigate this possibility, as suggested. We simulated the accumulation of neutral mutations for 24 h that would later confer the benefits of T, K, or G type mutations when appropriate Doxycycline- and/or Zeocin-containing conditions are simulated. Using this as the initial population, we ran the simulations with the parameter values that we previously found to approximate the experimental results. The types of mutations that arose in each condition did not change. We then asked what fraction of the final population of beneficial alleles originated

from these neutral mutants. We found that in conditions with strong selection and a monotonic fitness landscape (D2Z0 and D0Z2), these mutants comprised a notable fraction of final beneficial alleles (~35% and almost 50%, respectively). In all other conditions, they made up a very small fraction of the final population. We conclude that, in our system, readily available pre-existing mutants that later confer significant fitness advantages can, in fact, substantially contribute to the final beneficial mutant pool. However, rarer mutants (e.g., those required to fine-tune gene expression levels) are unlikely to pre-exist and contribute to the final mutant pool in exponentially growing cultures with periodic bottlenecks. We have updated the text to describe these findings in the last section of the manuscript.

C4. One possible outcome that hasn't been observed is rtTA mutating into tTA. tTA is the reverse of rtTA in that it activates a gene driven by TET02 promoter in the absence of doxycycline and represses the gene when doxycycline is present. These mutations are known in the literature. This can clearly rescue the cells' fitness in DiZ0 and D2Z0. The fact that this wasn't observed is puzzling. Some insight into why this wasn't observed would be helpful.

Response: This is an interesting possibility that could have indeed occurred considering a similar conversion of LacI into inverse LacI (see Poelwijk et al., Cell 146(3):462-70, 2011). In our case, it would be necessary for the “reverse TetR” or revTetR domain of rtTA to revert to TetR (such that it binds DNA in the absence of inducer). We cannot exclude the possibility of this happening – although we did not observe it in the clones that we tested. Overall, there are two reasons that suggest why the chance of this happening may be low. First, tTa would still contain the three intact VP16-derived F activator domains, and thus could maintain some toxicity even if it were to reverse its phenotype. The level of this toxicity is unknown because the fitness effects of bound and total tTA have not been characterized. If any tTA toxicity remains then cells with the tTA allele would be outcompeted by knockout “K” mutants that completely eliminate activator function and toxicity. Second, we checked that the revtetR sequence in our rtTA activator differs by 6 basepairs from the original tetR; see Urlinger et al., PNAS 97(14):7963-8 (2000). This suggests that although reversions to tTA by single basepair changes (different from these 6 basepairs) cannot be excluded, they may be unlikely. On the other hand, the reversion of all 6 basepairs to their original base in TetR is also very unlikely. Overall, we think that these two main reasons may make reversion to tTA a very rare event, difficult to observe in our experiments.

C5. The authors only plated and picked colonies of evolved strains from the 21st day (last day) of growth in each experiment (Figure 6). It's unclear that these strains actually represent the genotypes and phenotypes that exist in the populations during their most interesting recovery time (days 5-10 in all their evolution experiments). To get a sense for how many different strains and in what proportions the different strains are present within a single culture, the authors should consider plating the evolving cultures during the intense recovery phase (one of the days within Day 5 to Day 10) in which the growth rate really starts to go back up from the local minimum.

Response: We do have at least one whole-genome sequencing sample from days 5-10 in every experiment, so we definitely have some data on what genotypes were present during the main period of adaptation. In addition, we Sanger-sequenced and phenotypically characterized clones not only from day 21 but also from earlier time points in some environments. Specifically, we Sanger-sequenced individual colonies from the following time points:

12hr-DiZ0-r1:Day 13 (no mutation)

12hr-D2Z0-r1:Day 9 (rtTA+153: 3/12=25%, rtTA+123: 6/12=50%, rtTA+196: 8.3%, rtTA+410: 8.3%)

24hr-D2Z0-r1:Day 8 (rtTA+609: 20%, rtTA+95: 60%, rtTA+91: 10%, rtTA+454: 10%)

24hr-D2Z2-r1:Day 9 (rtTA+225: 50%, rtTA-9: 40%, rtTA+329: 10%)

24hr-D0Z2-r1:Day 9 (yEGFP::zeoR -219 Deletion: 8/10=80%, linked with GUP2 mutation for 2/10=20%)

In general, we found no linkage between mutations at these early time points, except in one case, where an intra-circuit mutation was linked to a GUP2 mutation in D0Z2. Therefore, we have some understanding of how the clones from later time points correspond to some mutations observed earlier in the experiment. Overall, we think that the situation is not unclear, and we hope that the phenotyping we performed will be sufficient to reveal the key aspects of evolutionary dynamics.

C6. Related to above, it's not clear to me that the "allele frequencies" obtained from sequencing the entire liquid culture (without isolating single colonies) at different days tells you about the fraction of different mutants present in a population. For example, in Fig. 3C & 3D, is the allele frequency telling us the fraction of cells within a population that has that allele? Does this method distinguish a scenario in which two mutations are present in the same cell from a scenario in which the two mutations are in two different cells? By plating liquid culture at different days (particularly at one of the days of sharp recovery: e.g., Day 5 & 6 in Replicate #2 in Fig. 3A), then picking individual colonies, sequencing them and characterizing their phenotypes, one could answer these types of questions. If I'm mistaken in my logic here, then the authors should more clearly explain what the allele frequency is telling us in terms of number of cell types that are present in one liquid culture over time.

Response: We thank the Reviewer for pointing out that we should have been clearer about this. We are really reporting read counts for Illumina and allele counts for Sanger sequencing data. We think, however, that the matter of linkage that the Reviewer mentions is a separate issue that we have addressed. Although the Illumina sequencing does not provide linkage information (at least not for the mutations we have, since they are generally more than one read length apart), we did isolate individual colonies at late time points and checked linkage with Sanger sequencing in D0Z2 and DiZ2 (Figs. 4B and 5E), exactly as the Reviewer suggested. Moreover, even with more sequencing we would expect minimal linkage at the beginning of the experiment due to the time required to accumulate multiple mutations.

C7. Figure 3: It's surprising that no mutations outside rtTA were observed even after 19 days. Were

there not even neutral mutations outside the circuit? How is the rate of neutral mutation inferred from Fig 3 (apparently very rare over 19 days) compatible with the seemingly more frequent rate of mutation in Fig. 4 & 5? This should be clearly explained in the text and the model.

Response: We think that the lack of extra-rtTA mutations can be understood in light of the overall mutation rate and selection on beneficial mutations. In general we believe that there is just not enough time for neutral mutations to fix in our experiment. This is because we only evolved the cells for a few hundred generations, while neutral mutations fix on the order of $N \sim 10^6$ generations according to theory. Therefore, neutral mutations should only exist at low frequencies or else must hitchhike linked to a beneficial mutation to become noticeable.

The difference between Fig. 3 (D2Z0) and Figs. 4 and 5 is more about the availability and selection on beneficial, rather than neutral mutations. In D2Z0, the selection on K mutations is so strong that they should easily outcompete any neutral and weakly beneficial mutants (such as T mutations). This is probably why we see only K-type mutations in the 21 days of the experiment. In D0Z2 and DiZ2, there seem to be many weakly-selected mutants, which is why we see several coexisting mutations for a longer period of time. However, we think that none of them are neutral.

C8. Comparing results from experiments in which dilution was done every 12-hours (Fig. 3-5) with those in which dilution was done every 24-hours (Extended view Figs. E3-E5), the mutations are quite different. For example, Fig. E3-B and Fig. 3D test the same D2Z0, but the mutations do not overlap in the two cases. Moreover, the population's growth rate seems to recover more quickly in the 24-hr resuspensions (in extended view) than in the 12-hr resuspensions (main figures). But the authors mention that they are basically the same. Please state more clearly how the dilution rate affects the evolution experiments.

Response: We thank the Reviewer for pointing this out. Indeed, while the outcomes of 12h and 24h resuspension experiments are similar in many ways, certain differences do exist between them. Similarities include: (i) the types of mutations and the genes where they occur (K mutations targeting rtTA in D2Z0, T mutations targeting rtTA in D2Z2 and G mutations in D0Z2); (ii) the ancestral genotype disappearing first in the same two conditions (D2Z0 and D0Z2); and (iii) the recurrence of mutations in the 225th basepair of rtTA in both conditions. Differences include the specific mutations in most conditions, and the speed at which the ancestral genotype is lost (apparently faster with 24-hour resuspensions).

C9. Related to point C3, the authors can stress more what's conceptually so important about the interplay between mutations that occur in their circuit and those occurring outside of the circuit. Can this be related to the time one would need to wait for a cell to find a beneficial mutation outside of the circuit? That is, what is the expected number of beneficial hits per unit time within the circuit and outside the circuit? Perhaps addressing this point gets to the heart of the trade-off between mutations

in and outside the circuit. If the authors can provide a conceptual insight on this point through their experiments or a model, it would significantly improve this manuscript.

Response: We agree that the interplay between circuit-internal and circuit-external mutations is an important aspect of this experiment. External mutations were observed only in two conditions: D0Z2 and DiZ2. We conclude that external beneficial mutations in other conditions are either not available or not sufficiently beneficial (and are therefore outcompeted by the internal mutations lowering rtTA toxicity). If their fitness effects are sufficiently large then external mutations will increase the number of potential ways of adaptation and thereby speed up evolution. Still, adaptation will not speed up linearly with the number of available mutations, because if the rate of beneficial mutations becomes very high, they will compete with each other, creating so-called “clonal interference”.

C10. One of messages that the authors stress is that they're studying how a "newcomer" network evolves. As they mention, it's true that a de novo network evolves quickly through horizontal gene transfer or recombination. But what is so special about being a newcomer isn't clear. Instead, they are really looking at a close network's evolution through mutations in itself and outside. There is no dotted line that says the circuit is new, and the cell certainly can't tell what is a "newcomer" circuit and what isn't. The only distinction that makes sense is whether you have a close network or not. So this focus on a "newcomer" network throughout the manuscript isn't clear to me.

Response: This is an interesting point. Networks can arise within genomes by gene duplication and recombination, but these would not be “newcomer” networks. The building material for such networks would be the existing genetic sequence. The PF network may still be considered as “newcomer” because its sequence is foreign to yeast. All coding sequences in the PF gene circuit are either from bacteria (tetR, zeoR), from jellyfish (yEGFP) or from viruses (VP16 activator domain). These sequences may have distinguishing features, such as not being codon-optimal in yeast. The network is completely human-designed and thereby new to the natural world. Nevertheless, we agree that calling the network “newcomer” does not add to the paper. Therefore, we removed this term from the revised version.

C11. The current presentation (both text and figures) detracts from what I think is a thoughtful and interesting study. The presentation would benefit from a substantial revision. Many of the plots in the main figures are not self-explanatory and not sufficiently explained in the figure captions. For example, it took me a long time just to figure out what the different colors or the numbers were in Fig. 3D and how those smooth lines were drawn there despite measuring only 3 time points (these are not explained in the captions, in the main text, or in the figure). The main text consists of many long sentences that contain unnecessary, and sometimes incorrectly applied technical jargon. Shortening these sentences, and using just simple basic words will clarify the authors' main message.

Response: Thank you for pointing this out. Regarding Fig. 3D and similar panels in other figures, we wrote a new section in the Experimental Procedures that explains how these plots were constructed. We added to the figure legends the following: “The way we used sequencing data to draw allele

frequencies and the lines connecting is explained in the Mutation time course reconstruction section of the Experimental Procedures.” **Likewise, we tried to edit the text carefully throughout the manuscript to clarify it and make it easier to read. We edited some of the figures and the figure legends. We hope that the revised manuscript is easier to digest.**

D. Minor comments

D1 - Many sentences were often too long and contained too many jargon. These made it difficult to understand the story at times. Often, no jargon would have been necessary to explain the experimental results.

Response: Thank you for pointing this out. We tried to edit the manuscript as suggested and eliminate long sentences and jargon as much as possible.

D2 - Figure 1B: Uses both "Cellular fitness" and "Population fitness". Cellular fitness is computed from the mathematical model. It's used to compute the population fitness. But the population fitness landscape in Fig. 1B should be measured, not calculated.

Response: We now dedicate a section in the Expanded View to explaining how the population fitness and gene expression distributions were measured in 13 different conditions, and how these data were used to calculate both cellular fitness and population fitness landscapes.

D3 - Figure 2A: Cartoon of dividing cells is confusing. Also, which category do mutations in ZeoR ORF fit in? K, T, and G seem only applicable to the rtTA or the TET promoters.

Response: We have updated the illustration in Fig. 2A. We hope that the revised version is easier to understand. The “G” mutation type includes ZeoR mutations – anything that elevates yEGFP::zeoR expression independently of rtTA is G type.

D4 - Figure 3A & 3B: The heat shades for the yEGFP measured in a flow cytometer. On some of the days (e.g., D2Z0 days 5-10 in replicate #1 in Fig. 3A), ON and OFF cells coexist in the same liquid culture. Is this because there are cells with bimodal expression (bistable) yEGFP or is it because there is a mixture of uni-modal ON and uni-modal OFF cells in the same liquid culture?

Response: This depends on the condition. In D2Z0, ancestral cells are fully expressing. When mutants appear, their expression will gradually decrease. Wild-type cells coexist temporarily with mutants in D2Z0, creating apparently bimodal expression patterns – but these are non-expressing and fully expressing genotypes in the same culture. On the other hand, in DiZ0 and DiZ2 mutant cells do maintain a bimodal distribution, which is different from the ancestral PF distribution.

D5 - Figure 3D: Measurements for only 3 of the days (Day 0, 9, 19) but smooth lines joining these points.

Not straight lines but lines with curvature. How did you get these lines? From a model or a spline-fitting? Moreover, the colors are not explained in the caption or in the figure.

Response: We have added a new section to the Experimental Procedures and have updated the figure captions to explain this.

D6 - Fig. 4B & 5D: Same comment as above.

Response: See above.

D7 - Fig. 6 is very confusing. It's unclear what the main message is here.

Response: We have previously tried multiple ways to represent the data shown in Fig. 6. So far, these bar graphs were found the easiest to digest – although we admit that they still require some thought and interpretation. We have tried to add clarifications to the manuscript and the legend, hoping that this figure is now somewhat easier to interpret. Essentially, the goal of the figure is to show how the gene expression and fitness of various evolved clonal isolates compares to the ancestral PF cells. For clones evolved in D2Z0 and D0Z2 we perform these comparisons only in the conditions where they evolved - simply testing if rtTA has lost toxicity and if the cells are Zeocin-resistant. However, for clones evolved in D2Z2 and DiZ2, the evolved phenotypes are more complex. To capture this, we perform comparisons in many different conditions, and consequently the corresponding plots may be more difficult to interpret.

D8 - Extended view - Fig.E3-B doesn't look qualitatively the same as Fig. 3D, even though they are for the same environmental condition: D2Z0.

Response: This is indeed true and the reasons are not entirely clear. Apparently, evolution is faster with 24-hour resuspensions compared to 12-hour resuspensions.

Thank you again for submitting your work to Molecular Systems Biology. We have now heard back from the referee who agreed to evaluate your manuscript. As you will see below, the referee is now satisfied with the modifications made and thinks that the study is now suitable for publication.

Before formally accepting the manuscript, we would ask you to address some minor editorial issues listed below.

Reviewer #2:

Summary:

I recommend publication.

The authors have addressed most of my comments. They also seem to have addressed most of Reviewer 1's comments. The revised figures and text have significantly improved the clarity. It's still a very "information dense" paper but I think the authors have done a very good job in improving it from the previous version. It clarified some points that I previously misunderstood.

Below are my comments on the authors' responses to the "major comments" that I listed in my first evaluation report:

C1. Satisfied that the previous emphasis on the bet-hedging angle has now been eliminated. Satisfied with the authors' statistical analysis of DiZ0 vs D0Z0 conditions.

C2. Satisfied that the model is now more clearly explained than in the previous version.

C3. Satisfied with the inclusion of new analyses on pre-existing mutants.

C4. I think this is a likely explanation.

C5-C6: The authors clarified my confusion regarding the sequencing data.

C7: Satisfied.

C8. Satisfied.

C9. Satisfied.

C10. Satisfied with the removal of the "newcomer" in the title and text.

C11. Satisfied.